# Testing theory of mind in large language models and humans

**James W. A. Strachan** [1] ✉, **Dalila Albergo** [2,3], **Giulia Borghini**[2],
**Oriana Pansardi** [1,2,4], **Eugenio Scaliti** [1,2,5,6], **Saurabh Gupta** [7], **Krati Saxena** [7],
**Alessandro Rufo** [7], **Stefano Panzeri** [8], **Guido Manzi** [7],
**Michael S. A. Graziano**[9] **& Cristina Becchio** [1,2] ✉

At the core of what defines us as humans is the concept of theory of mind: the ability to track other people's mental states. The recent development of large language models (LLMs) such as ChatGPT has led to intense debate about the possibility that these models exhibit behaviour that is indistinguishable from human behaviour in theory of mind tasks. Here we compare human and LLM performance on a comprehensive battery of measurements that aim to measure different theory of mind abilities, from understanding false beliefs to interpreting indirect requests and recognizing irony and faux pas. We tested two families of LLMs (GPT and LLaMA2) repeatedly against these measures and compared their performance with those from a sample of 1,907 human participants. Across the battery of theory of mind tests, we found that GPT-4 models performed at, or even sometimes above, human levels at identifying indirect requests, false beliefs and misdirection, but struggled with detecting faux pas. Faux pas, however, was the only test where LLaMA2 outperformed humans. Follow-up manipulations of the belief likelihood revealed that the superiority of LLaMA2 was illusory, possibly reflecting a bias towards attributing ignorance. By contrast, the poor performance of GPT originated from a hyperconservative approach towards committing to conclusions rather than from a genuine failure of inference. These findings not only demonstrate that LLMs exhibit behaviour that is consistent with the outputs of mentalistic inference in humans but also highlight the importance of systematic testing to ensure a non-superficial comparison between human and artificial intelligences.

People care about what other people think and expend a lot of effort thinking about what is going on in other minds. Everyday life is full of social interactions that only make sense when considered in light of our capacity to represent other minds: when you are standing near a closed window and a friend says, 'It's a bit hot in here', it is your ability to think about her beliefs and desires that allows you to recognize that she is not just commenting on the temperature but politely asking you to open the window[1].

[1]Department of Neurology, University Medical Center Hamburg-Eppendorf, Hamburg, Germany. [2]Cognition, Motion and Neuroscience, Italian Institute of Technology, Genoa, Italy. [3]Center for Mind/Brain Sciences, University of Trento, Rovereto, Italy. [4]Department of Psychology, University of Turin, Turin, Italy. [5]Department of Management, 'Valter Cantino', University of Turin, Turin, Italy. [6]Human Science and Technologies, University of Turin, Turin, Italy. [7]Alien Technology Transfer Ltd, London, UK. [8]Institute for Neural Information Processing, Center for Molecular Neurobiology, University Medical Center Hamburg- Eppendorf, Hamburg, Germany. [9]Princeton Neuroscience Institute, Princeton University, Princeton, NJ, USA.
✉e-mail: james.wa.strachan@gmail.com; c.becchio@uke.de

This ability for tracking other people's mental states is known as theory of mind. Theory of mind is central to human social interactions—from communication to empathy to social decision-making—and has long been of interest to developmental, social and clinical psychologists. Far from being a unitary construct, theory of mind refers to an interconnected set of notions that are combined to explain, predict, and justify the behaviour of others[2]. Since the term 'theory of mind' was first introduced in 1978 (ref. [3]), dozens of tasks have been developed to study it, including indirect measures of belief attribution using reaction times[4–6] and looking or searching behaviour[7–9], tasks examining the ability to infer mental states from photographs of eyes[10], and language-based tasks assessing false belief understanding[11,12] and pragmatic language comprehension[13–16]. These measures are proposed to test early, efficient but inflexible implicit processes as well as later-developing, flexible and demanding explicit abilities that are crucial for the generation and comprehension of complex behavioural interactions[17,18] involving phenomena such as misdirection, irony, implicature and deception.

The recent rise of large language models (LLMs), such as generative pre-trained transformer (GPT) models, has shown some promise that artificial theory of mind may not be too distant an idea. Generative LLMs exhibit performance that is characteristic of sophisticated decision-making and reasoning abilities[19,20] including solving tasks widely used to test theory of mind in humans[21–24]. However, the mixed success of these models[23], along with their vulnerability to small perturbations to the provided prompts, including simple changes in characters' perceptual access[25], raises concerns about the robustness and interpretability of the observed successes. Even in cases where these models are capable of solving complex tasks[20] that are cognitively demanding even for human adults[17], it cannot be taken for granted that they will not be tripped up by a simpler task that a human would find trivial[26]. As a result, work in LLMs has begun to question whether these models rely on shallow heuristics rather than robust performance that parallels human theory of mind abilities[27].

In the service of the broader multidisciplinary study of machine behaviour[28], there have been recent calls for a 'machine psychology'[29] that have argued for using tools and paradigms from experimental psychology to systematically investigate the capacities and limits of LLMs[30]. A systematic experimental approach to studying theory of mind in LLMs involves using a diverse set of theory of mind measures, delivering multiple repetitions of each test, and having clearly defined benchmarks of human performance against which to compare[31]. In this Article, we adopt such an approach to test the performance of LLMs in a wide range of theory of mind tasks. We tested the chat-enabled version of GPT-4, the latest LLM in the GPT family of models, and its predecessor ChatGPT-3.5 (hereafter GPT-3.5) in a comprehensive set of psychological tests spanning different theory of mind abilities, from those that are less cognitively demanding for humans such as understanding indirect requests to more cognitively demanding abilities such as recognizing and articulating complex mental states like misdirection or irony[17]. GPT models are closed, evolving systems. In the interest of reproducibility[32], we also tested the open-weight LLaMA2-Chat models on the same tests. To understand the variability and boundary limitations of LLMs' social reasoning capacities, we exposed each model to multiple repetitions of each test across independent sessions and compared their performance with that of a sample of human participants (total $N = 1,907$). Using variants of the tests considered, we were able to examine the processes behind the models' successes and failures in these tests.

## Results

### Theory of mind battery

We selected a set of well-established theory of mind tests spanning different abilities: the hinting task[14], the false belief task[11,33], the recognition of faux pas[13], and the strange stories[15,16]. We also included a test of irony comprehension using stimuli adapted from a previous study[34]. Each test was administered separately to GPT-4, GPT-3.5 and LLaMA2-70B-Chat (hereafter LLaMA2-70B) across 15 chats. We also tested two other sizes of LLaMA2 model (7B and 13B), the results of which are reported in Supplementary Information section 1. Because each chat is a separate and independent session, and information about previous sessions is not retained, this allowed us to treat each chat (session) as an independent observation. Responses were scored in accordance with the scoring protocols for each test in humans (Methods) and compared with those collected from a sample of 250 human participants. Tests were administered by presenting each item sequentially in a written format that ensured a species-fair comparison[35] (Methods) between LLMs and human participants.

### Performance across theory of mind tests

Except for the irony test, all other tests in our battery are publicly available tests accessible within open databases and scholarly journal articles. To ensure that models did not merely replicate training set data, we generated novel items for each published test (Methods). These novel test items matched the logic of the original test items but used a different semantic content. The text of original and novel items and the coded responses are available on the OSF (methods and resource availability).

Figure 1a compares the performance of LLMs against the performance of human participants across all tests included in the battery. Differences in performance on original items versus novel items, separately for each test and model, are shown in Fig. 1b.

**False belief.** Both human participants and LLMs performed at ceiling on this test (Fig. 1a). All LLMs correctly reported that an agent who left the room while the object was moved would later look for the object in the place where they remembered seeing it, even though it no longer matched the current location. Performance on novel items was also near perfect (Fig. 1b), with only 5 human participants out of 51 making one error, typically by failing to specify one of the two locations (for example, 'He'll look in the room'; Supplementary Information section 2).

In humans, success on the false belief task requires inhibiting one's own belief about reality in order to use one's knowledge about the character's mental state to derive predictions about their behaviour. However, with LLMs, performance may be explained by lower-level explanations than belief tracking[27]. Supporting this interpretation, LLMs such as ChatGPT have been shown to be susceptible to minor alterations to the false belief formulation[25,27], such as making the containers where the object is hidden transparent or asking about the belief of the character who moved the object rather than the one who was out of the room. Such perturbations of the standard false belief structure are assumed not to matter for humans (who possess a theory of mind)[25]. In a control study using these perturbation variants (Supplementary Information section 4 and Supplementary Appendix 1), we replicated the poor performance of GPT models found in previous studies[25]. However, we found that human participants ($N = 757$) also failed on half of these perturbations. Understanding these failures and the similarities and differences in how humans and LLMs may arrive at the same outcome requires further systematic investigation. For example, because these perturbations also involve changes in the physical properties of the environment, it is difficult to establish whether LLMs (and humans) failed because they were sticking to the familiar script and were unable to automatically attribute an updated belief, or because they did not consider physical principles (for example, transparency).

**Irony.** GPT-4 performed significantly better than human levels ($Z = 0.00$, $P = 0.040$, $r = 0.32$, 95% confidence interval (CI) 0.14−0.48). By contrast, both GPT-3.5 ($Z = −0.17$, $P = 2.37 \times 10^{-6}$, $r = 0.64$, 95% CI 0.49−0.77) and LLaMA2-70B ($Z = −0.42$, $P = 2.39 \times 10^{-7}$, $r = 0.70$, 95% CI 0.55−0.79) performed below human levels (Fig. 1a). GPT-3.5 performed perfectly at

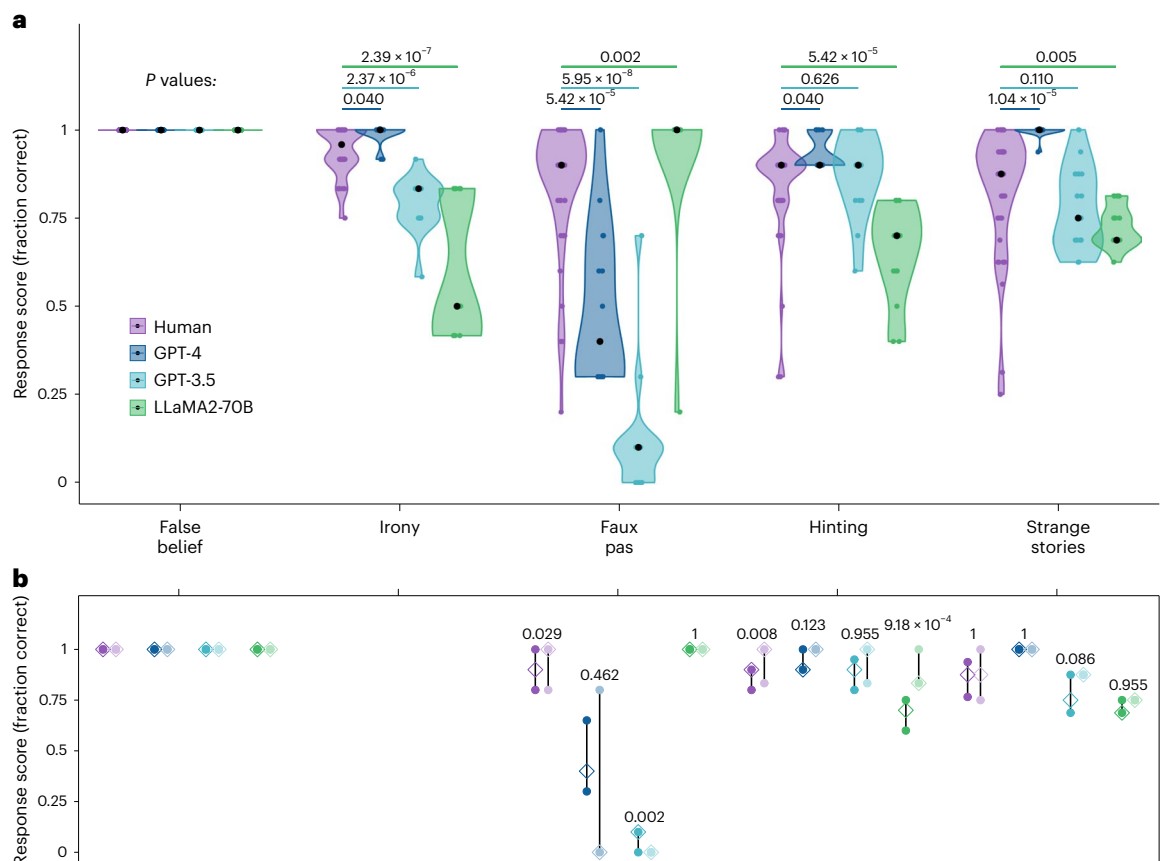

**Fig. 1 | Performance of human (purple), GPT-4 (dark blue), GPT-3.5 (light blue) and LLaMA2-70B (green) on the battery of theory of mind tests. a**, Original test items for each test showing the distribution of test scores for individual sessions and participants. Coloured dots show the average response score across all test items for each individual test session (LLMs) or participant (humans). Black dots indicate the median for each condition. $P$ values were computed from Holm-corrected Wilcoxon two-way tests comparing LLM scores ($n = 15$ LLM observations) against human scores (irony, $N = 50$ human participants; faux pas, $N = 51$ human participants; hinting, $N = 48$ human participants; strange stories, $N = 50$ human participants). Tests are ordered in descending order of human performance. **b**, Interquartile ranges of the average scores on the original published items (dark colours) and novel items (pale colours) across each test (for LLMs, $n = 15$ LLM observations; for humans, false belief, $N = 49$ human participants; faux pas, $N = 51$ human participants; hinting, $N = 48$ human participants; strange stories, $N = 50$ human participants). Empty diamonds indicate the median scores, and filled circles indicate the upper and lower bounds of the interquartile range. $P$ values shown are from Holm-corrected Wilcoxon two-way tests comparing performance on original items against the novel items generated as controls for this study.

recognizing non-ironic control statements but made errors at recognizing ironic utterances (Supplementary Information section 2). Control analysis revealed a significant order effect, whereby GPT-3.5 made more errors on earlier trials than later ones (Supplementary Information section 3). LLaMA2-70B made errors when recognizing both ironic and non-ironic control statements, suggesting an overall poor discrimination of irony.

**Faux Pas.** On this test, GPT-4 scored notably lower than human levels ($Z = -0.40$, $P = 5.42 \times 10^{-5}$, $r = 0.55$, 95% CI 0.33–0.71) with isolated ceiling effects on specific items (Supplementary Information section 2). GPT-3.5 scored even worse, with its performance nearly at floor ($Z = -0.80$, $P = 5.95 \times 10^{-8}$, $r = 0.72$, 95% CI 0.58–0.81) on all items except one. By contrast, LLaMA2-70B outperformed humans ($Z = 0.10$, $P = 0.002$, $r = 0.44$, 95% CI 0.24–0.61) achieving 100% accuracy in all but one run.

The pattern of results for novel items was qualitatively similar (Fig. 1b). Compared with original items, the novel items proved slightly easier for humans ($Z = -0.10$, $P = 0.029$, $r = 0.29$, 95% CI 0.10–0.50) and more difficult for GPT-3.5 ($Z = 0.10$, $P = 0.002$, $r = 0.69$, 95% CI 0.49–0.88), but not for GPT-4 and LLaMA2-70B ($P > 0.462$; Bayes factor ($BF_{10}$) of 0.77 and 0.43, respectively). Given the poor performance of GPT-3.5 of the original test items, this difference was unlikely to be explained by a prior familiarity with the original items. These results were robust to alternative coding schemes (Supplementary Information section 5).

**Hinting.** On this test, GPT-4 performance was significantly better than humans ($Z = 0.00$, $P = 0.040$, $r = 0.32$, 95% CI 0.12–0.50). GPT-3.5 performance did not significantly differ from human performance ($Z = 0.00$, $P = 0.626$, $r = 0.06$, 95% CI 0.01–0.33, $BF_{10}$ 0.33). Only LLaMA2-70B scored significantly below human levels of performance on this test ($Z = -0.20$, $P = 5.42 \times 10^{-5}$, $r = 0.57$, 95% CI 0.41–0.72).

Novel items proved easier than original items for both humans ($Z = -0.10$, $P = 0.008$, $r = 0.34$, 95% CI 0.14–0.53) and LLaMA2-70B ($Z = -0.20$, $P = 9.18 \times 10^{-4}$, $r = 0.73$, 95% CI 0.50–0.87) (Fig. 1b). Scores on novel items did not differ from the original test items for GPT-3.5 ($Z = -0.03$, $P = 0.955$, $r = 0.24$, 95% CI 0.02–0.59, $BF_{10}$ 0.61) or GPT-4 ($Z = -0.10$, $P = 0.123$, $r = 0.44$, 95% CI 0.07–0.75, $BF_{10}$ 0.91). Given that better performance on novel items is the opposite of what a prior familiarity explanation would predict, it is likely that this difference for LLaMA2-70B was driven by differences in item difficulty.

**Strange stories.** GPT-4 significantly outperformed humans on this test ($Z = 0.13$, $P = 1.04 \times 10^{-5}$, $r = 0.60$, 95% CI 0.46–0.72). The performance of GPT-3.5 did not significantly differ from humans ($Z = -0.06$, $P = 0.110$, $r = 0.24$, 95% CI 0.03–0.44, $BF_{10}$ 0.47), while LLaMA2-70B scored significantly lower than humans ($Z = -0.13$, $P = 0.005$, $r = 0.41$, 95% CI 0.24–0.60). There were no differences between original and novel items for any model (all $P > 0.085$; $BF_{10}$: human 0.22, GPT-3.5 1.46, LLaMA2-70B 0.46; the variance for GPT-4 was too low to compute

a Bayes factor). As reported in Supplementary Information section 6, partial successes were infrequent and more likely for LLaMA2-70B than for other models.

## Understanding faux pas

In line with previous findings that GPT models struggle with faux pas[36], in our battery, faux pas was the only test in which GPT-4 did not match or exceed human performance. Surprisingly, faux pas was also the only test in which LLaMA2-70B, which was otherwise the poorest-performing model, scored better than humans (Fig. 1).

The faux pas test consists of vignettes describing an interaction where one character (the speaker) says something they should not have said, not knowing or not realizing that they should not say it. To understand that a faux pas has occurred, one must represent two mental states: that the speaker does not know that they should not say what they said, and that the person hearing it (the victim) would feel insulted or hurt by what was said. Consider the following example of a story describing a faux pas:

```
Jill had just moved into a new house. She went shop-
ping with her Mum and bought some new curtains. When
Jill had just put them up, her best friend Lisa came
round and said, "Oh, those curtains are horrible, I
hope you're going to get some new ones." Jill asked,
"Do you like the rest of my bedroom?"
```

Following the presentation of the story, four comprehension questions are asked. The first question is, 'In the story, did somebody say something they should not have said?' to which the correct answer is always yes. The second question asks the respondent to report what the person said that they should not have said, and the third question is a comprehension question specific to the content of the story. The fourth and key question, which was our focus for coding, relates to the speaker's belief when they made the inappropriate utterance: 'Did [Lisa] know that [the curtains were new]?' The correct answer to this final question is always no.

Both GPT-4 and GPT-3.5 correctly identified that the victim would feel insulted or hurt, sometimes going so far as to provide additional details about why the utterance might cause offence. However, when asked whether the speaker was aware of the context that made their statement offensive (such as Lisa knowing that Jill had just bought the curtains), they failed to produce a correct answer. A closer look revealed that the overwhelming majority of the errors on this question reported that there was not enough information provided to be sure, for example:

```
Did Lisa know the curtains were new?
GPT-4: […] It is unclear from the story whether Lisa
knew the curtains were new or not.
```

Only two responses out of 349 reported that, yes, the character did know. We consider three alternative hypotheses for why GPT models, and specifically GPT-4, fail to answer this question correctly.

The first hypothesis, which we term the failure of inference hypothesis, is that models fail to generate inferences about the mental state of the speaker (note that we refer to inference here not in the sense of the processes by which biological organisms infer hidden states from their environment, but rather as any process of reasoning whereby conclusions are derived from a set of propositional premises). Recognizing a faux pas in this test relies on contextual information beyond that encoded within the story (for example, about social norms). For example, in the above example there is no information in the story to indicate that saying that the newly bought curtains are horrible is inappropriate, but this is a necessary proposition that must be accepted in order to accurately infer the mental states of the characters. This

inability to use non-embedded information would fundamentally impair the ability of GPT-4 to compute inferences.

The second hypothesis, which we term the Buridan's ass hypothesis, is that models are capable of inferring mental states but cannot choose between them, as with the eponymous rational agent caught between two equally appetitive bales of hay that starves because it cannot resolve the paradox of making a decision in the absence of a clear preference[37]. Under this hypothesis, GPT models can propose the correct answer (a faux pas) as one among several possible alternatives but do not rank these alternatives in terms of likelihood. In partial support of this hypothesis, responses from both GPT models occasionally indicate that the speaker may not know or remember but present this as one hypothesis among alternatives (Supplementary Information section 5).

The third hypothesis, which we term the hyperconservatism hypothesis, is that GPT models are able both to compute inferences about the mental states of characters and recognise a false belief or lack of knowledge as the likeliest explanation among competing alternatives but refrain from committing to a single explanation out of an excess of caution. GPT models are powerful language generators, but they are also subject to inhibitory mitigation processes[38]. It is possible that such processes could lead to an overly conservative stance where GPT models do not commit to the likeliest explanation despite being able to generate it.

To differentiate between these hypotheses, we devised a variant of the faux pas test where the question assessing performance on the faux pas test was formulated in terms of likelihood (hereafter, the faux pas likelihood test). Specifically, rather than ask whether the speaker knew or did not know, we asked whether it was more likely that the speaker knew or did not know. Under the hyperconservatism hypothesis, GPT models should be able to both make the inference that the speaker did not know and identify it as more likely among alternatives, and so we would expect the models to respond accurately that it was more likely that the speaker did not know. In case of uncertainty or incorrect responses, we further prompted models to describe the most likely explanation. Under the Buridan's ass hypothesis, we expected this question would elicit multiple alternative explanations that would be presented as equally plausible, while under the failure of inference hypothesis, we expected that GPT would not be able to generate the right answer at all as a plausible explanation.

As shown in Fig. 2a, on the faux pas likelihood test GPT-4 demonstrated perfect performance, with all responses identifying without any prompting that it was more likely that the speaker did not know the context. GPT-3.5 also showed improved performance, although it did require prompting in a few instances (~3% of items) and occasionally failed to recognize the faux pas (~9% of items; see Supplementary Information section 7 for a qualitative analysis of response types).

Taken together, these results support the hyperconservatism hypothesis, as they indicate that GPT-4, and to a lesser but still notable extent GPT-3.5, successfully generated inferences about the mental states of the speaker and identified that an unintentional offence was more likely than an intentional insult. Thus, failure to respond correctly to the original phrasing of the question does not reflect a failure of inference, nor indecision among alternatives the model considered equally plausible, but an overly conservative approach that prevented commitment to the most likely explanation.

## Testing information integration

A potential confound of the above results is that, as the faux pas test includes only items where a faux pas occurs, any model biased towards attributing ignorance would demonstrate perfect performance without having to integrate the information provided by the story. This potential bias could explain the perfect performance of LLaMA2-70B in the original faux pas test (where the correct answer is always, 'no') as well as GPT-4's perfect and GPT-3.5's good performance on the faux

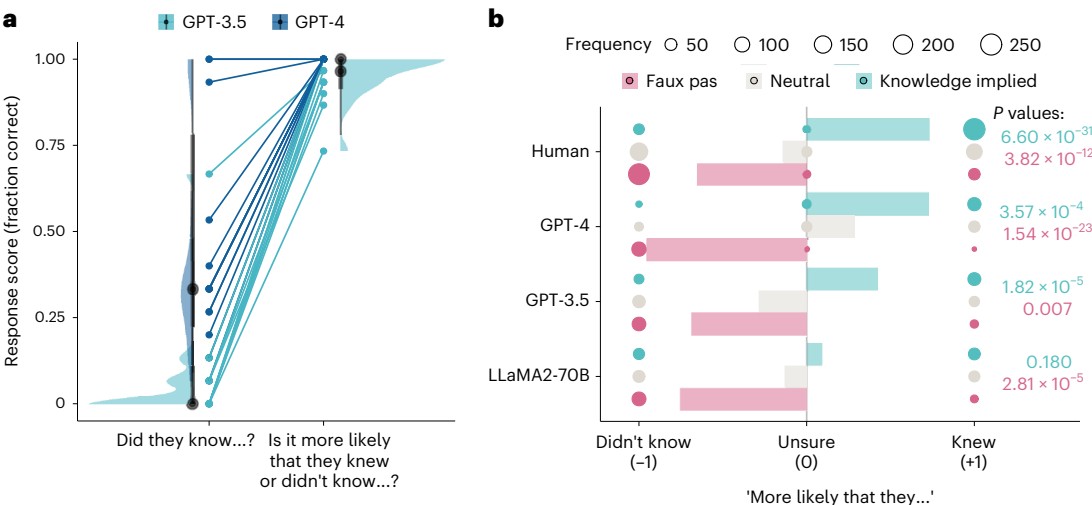

**Fig. 2 | Results of the variants of the faux pas test. a**, Scores of the two GPT models on the original framing of the faux pas question ('Did they know…?') and the likelihood framing ('Is it more likely that they knew or didn't know…?'). Dots show average score across trials ($n = 15$ LLM observations) on particular items to allow comparison between the original faux pas test and the new faux pas likelihood test. Halfeye plots show distributions, medians (black points), 66% (thick grey lines) and 99% quantiles (thin grey lines) of the response scores on different items ($n = 15$ different stories involving faux pas). **b**, Response scores to three variants of the faux pas test: faux pas (pink), neutral (grey) and knowledge-implied variants (teal). Responses were coded as categorical data as 'didn't know', 'unsure' or 'knew' and assigned a numerical coding of −1, 0 and +1. Filled balloons are shown for each model and variant, and the size of each balloon indicates the count frequency, which was the categorical data used to compute chi-square tests. Bars show the direction bias score computed as the average across responses of the categorical data coded as above. On the right of the plot, $P$ values (one-sided) of Holm-corrected chi-square tests are shown comparing the distribution of response type frequencies in the faux pas and knowledge-implied variants against neutral.

pas likelihood test (where the correct answer is always 'more likely that they didn't know').

To control for this, we developed a novel set of variants of the faux pas likelihood test manipulating the likelihood that the speaker knew or did not know (hereafter the belief likelihood test). For each test item, all newly generated for this control study, we created three variants: a 'faux pas' variant, a 'neutral' variant, and a 'knowledge-implied' variant (Methods). In the faux pas variant, the utterance suggested that the speaker did not know the context. In the neutral variant, the utterance suggested neither that they knew nor did not know. In the knowledge-implied variant, the utterance suggested that the speaker knew (for the full text of all items, see Supplementary Appendix 2).

If the models' responses reflect a true discrimination of the relative likelihood of the two explanations (that the person knew versus that they didn't know, hereafter 'knew' and 'didn't know'), then the distribution of 'knew' and 'didn't know' responses should be different across variants. Specifically, relative to the neutral variant, 'didn't know' responses should predominate for the faux pas, and 'knew' responses should predominate for the knowledge-implied variant. If the responses of the models do not discriminate between the three variants, or discriminate only partially, then it is likely that responses are affected by a bias or heuristic unrelated to the story content.

We adapted the three variants (faux pas, neutral and knowledge implied) for six stories, administering each test item separately to each LLM and a new sample of human participants (total $N = 900$). Responses were coded using a numeric code to indicate which, if either, of the knew/didn't know explanations the response endorsed (−1, didn't know; 0, unsure or impossible to tell; +1, knew). These coded scores were then averaged for each story to give a directional score for each variant such that negative values indicated the model was more likely to endorse the 'didn't know' explanation, while positive values indicated the model was more likely to endorse the 'knew' explanation. These results are shown in Fig. 2b. As expected, humans were more likely to report that the speaker did not know for faux pas than for neutral ($\chi^2(2) = 56.20$, $P = 3.82 \times 10^{-12}$) and more likely to report that the speaker did know

for knowledge implied than for neutral ($\chi^2(2) = 143$, $P = 6.60 \times 10^{-31}$). Humans also reported uncertainty on a small proportion of trials, with a higher proportion in the neutral condition (28 out of 303 responses) than in the other variants (11 out of 303 for faux pas, and 0 out of 298 for knowledge implied).

Similarly to humans, GPT-4 was more likely to endorse the 'didn't know' explanation for faux pas than for neutral ($\chi^2(2) = 109$, $P = 1.54 \times 10^{-23}$) and more likely to endorse the 'knew' explanation for knowledge implied than for neutral ($\chi^2(2) = 18.10$, $P = 3.57 \times 10^{-4}$). GPT-4 was also more likely to report uncertainty in the neutral condition than responding randomly (42 out of 90 responses, versus 6 and 17 in the faux pas and knowledge-implied variants, respectively).

The pattern of responses for GPT-3.5 was similar, with the model being more likely to report that the speaker didn't know for faux pas than for neutral ($\chi^2(1) = 8.44$, $P = 0.007$) and more likely that the character knew for knowledge implied than for neutral ($\chi^2(1) = 21.50$, $P = 1.82 \times 10^{-5}$). Unlike GPT-4, GPT-3.5 never reported uncertainty in response to any variants and always selected one of the two explanations as the likelier even in the neutral condition.

LLaMA2-70B was also more likely to report that the speaker didn't know in response to faux pas than neutral ($\chi^2(1) = 20.20$, $P = 2.81 \times 10^{-5}$), which was consistent with this model's ceiling performance in the original formulation of the test. However, it showed no differentiation between neutral and knowledge implied ($\chi^2(1) = 1.80$, $P = 0.180$, $BF_{10}$ 0.56). As with GPT-3.5, LLaMA2-70B never reported uncertainty in response to any variants and always selected one of the two explanations as the likelier.

Furthermore, the responses of LLaMA2-70B and, to a lesser extent, GPT-3.5 appeared to be subject to a response bias towards affirming that someone had said something they should not have said. Although the responses to the first question (which involved recognising that there was an offensive remark made) were of secondary interest to our study, it was notable that, although all models could correctly identify that an offensive remark had been made in the faux pas condition (all LLMs 100%, humans 83.61%), only GPT-4 reliably reported that there was no offensive statement in the neutral and knowledge-implied conditions (15.47% and 27.78%, respectively), with similar proportions to human

responses (neutral 19.27%, knowledge implied 30.10%). GPT-3.5 was more likely to report that somebody made an offensive remark in all conditions (neutral 71.11%, knowledge implied 87.78%), and LLaMA2-70B always reported that somebody in the story had made an offensive remark.

## Discussion

We collated a battery of tests to comprehensively measure performance in theory of mind tasks in three LLMs (GPT-4, GPT-3.5 and LLaMA2-70B) and compared these against the performance of a large sample of human participants. Our findings validate the methodological approach taken in this study using a battery of multiple tests spanning theory of mind abilities, exposing language models to multiple sessions and variations in both structure and content, and implementing procedures to ensure a fair, non-superficial comparison between humans and machines[35]. This approach enabled us to reveal the existence of specific deviations from human-like behaviour that would have remained hidden using a single theory of mind test, or a single run of each test.

Both GPT models exhibited impressive performance in tasks involving beliefs, intentions and non-literal utterances, with GPT-4 exceeding human levels in the irony, hinting and strange stories. Both GPT-4 and GPT-3.5 failed only on the faux pas test. Conversely, LLaMA2-70B, which was otherwise the poorest-performing model, outperformed humans on the faux pas. Understanding a faux pas involves two aspects: recognizing that one person (the victim) feels insulted or upset and understanding that another person (the speaker) holds a mistaken belief or lacks some relevant knowledge. To examine the nature of models' successes and failures on this test, we developed and tested new variants of the faux pas test in a set of control experiments.

Our first control experiment using a likelihood framing of the belief question (faux pas likelihood test), showed that GPT-4, and to a lesser extent GPT-3.5, correctly identified the mental state of both the victim and the speaker and selected as the most likely explanation the speaker not knowing or remembering the relevant knowledge that made their statement inappropriate. Despite this, both models consistently provided an incorrect response (at least when compared against human responses) when asked whether the speaker knew or remembered this knowledge, responding that there was insufficient information provided. In line with the hyperconservatism hypothesis, these findings imply that, while GPT models can identify unintentional offence as the most likely explanation, their default responses do not commit to this explanation. This finding is consistent with longitudinal evidence that GPT models have become more reluctant to answer opinion questions over time[39].

Further supporting that the failures of GPT at recognizing faux pas were due to hyperconservatism in answering the belief question rather than a failure of inference, a second experiment using the belief likelihood test showed that GPT responses integrated information in the story to accurately interpret the speaker's mental state. When the utterance suggested that the speaker knew, GPT responses acknowledged the higher likelihood of the 'knew' explanation. LLaMA2-70B, on the other hand, did not differentiate between scenarios where the speaker was implied to know and when there was no information one way or another, raising the concern that the perfect performance of LLaMA2-70B on this task may be illusory.

The pattern of failures and successes of GPT models on the faux pas test and its variants may be the result of their underlying architecture. In addition to transformers (generative algorithms that produce text output), GPT models also include mitigation measures to improve factuality and avoid users' overreliance on them as sources[38]. These measures include training to reduce hallucinations, the propensity of GPT models to produce nonsensical content or fabricate details that are not true in relation to the provided content. Failure on the faux pas test may be an exercise of caution driven by these mitigation measures,

as passing the test requires committing to an explanation that lacks full evidence. This caution can also explain differences between tasks: both the faux pas and hinting tests require speculation to generate correct answers from incomplete information. However, while the hinting task allows for open-ended generation of text in ways to which LLMs are well suited, answering the faux pas test requires going beyond this speculation in order to commit to a conclusion.

The cautionary epistemic policy guiding the responses of GPT models introduces a fundamental difference in the way that humans and GPT models respond to social uncertainty[40]. In humans, thinking is, first and last, for the sake of doing[41,42]. Humans generally find uncertainty in social environments to be aversive and will incur additional costs to reduce it[43]. Theory of mind is crucial in reducing such uncertainty; the ability to reason about mental states—in combination with information about context, past experience and knowledge of social norms—helps individual reduce uncertainty and commit to likely hypotheses, allowing for successful navigation of the social environment as active agents[44,45]. GPT models, on the other hand, respond conservatively despite having access to tools to reduce uncertainty. The dissociation we describe between speculative reasoning and commitment mirrors recent evidence that, while GPT models demonstrate sophisticated and accurate performance in reasoning tasks about belief states, they struggle to translate this reasoning into strategic decisions and actions[46].

These findings highlight a dissociation between competence and performance[35], suggesting that GPT models may be competent, that is, have the technical sophistication to compute mentalistic-like inferences but perform differently from humans under uncertain circumstances as they do not compute these inferences spontaneously to reduce uncertainty. Such a distinction can be difficult to capture with quantitative approaches that code only for target response features, as machine failures and successes are the result of non-human-like processes[30] (see Supplementary Information section 7 for a preliminary qualitative breakdown of how GPT models' successes on the new version of the faux pas test may not necessarily reflect perfect or human-like reasoning).

While LLMs are designed to emulate human-like responses, this does not mean that this analogy extends to the underlying cognition giving rise to those responses[47]. In this context, our findings imply a difference in how humans and GPT models trade off the costs associated with social uncertainty against the costs associated with prolonged deliberation[48]. This difference is perhaps not surprising considering that resolving uncertainty is a priority for brains adapted to deal with embodied decisions, such as deciding whether to approach or avoid, fight or flight, or cooperate or defect. GPT models and other LLMs do not operate within an environment and are not subject to the processing constraints that biological agents face to resolve competition between action choices, so may have limited advantages in narrowing the future prediction space[46,49,50].

The dis-embodied cognition of GPT models could explain failures in recognizing faux pas, but they may also underlie their success on other tests. One example is the false belief test, one of the most widely used tools so far for testing the performance of LLMs on social cognitive tasks[19,21–23,25,51,52]. In this test, participants are presented with a story where a character's belief about the world (the location of the item) differs from the participant's own belief. The challenge in these stories is not remembering where the character last saw the item but rather in reconciling the incongruence between conflicting mental states. This is challenging for humans, who have their own perspective, their own sense of self and their own ability to track out-of-sight objects. However, if a machine does not have its own self-perspective because it is not subject to the constraints of navigating a body through an environment, as with GPT[53], then tracking the belief of a character in a story does not pose the same challenge.

An important direction for future research will be to examine the impact of these non-human decision behaviours on second-person,

**Table 1 | Data collection details for each model**

| Test | Model | N/n | Items | Dates of data collection |
|---|---|---|---|---|
| Theory of mind battery | Human | 250 | 7–16 | June to July 2023 |
| | GPT-4 | 75 | 7–16 | April 2023 |
| | GPT-3.5 | 75 | 7–16 | April 2023 |
| | LLaMA2 | 75 | 7–16 | October to November 2023 |
| Faux pas likelihood test | GPT-4 | 15 | 15 | April to May 2023 |
| | GPT-3.5 | 15 | 15 | April to May 2023 |
| | LLaMA2 | 15 | 15 | October to November 2023 |
| Belief likelihood test | Human | 900 | 1 | November 2023 |
| | GPT-4 | 270 | 1 | October to November 2023 |
| | GPT-3.5 | 270 | 1 | October to November 2023 |
| | LLaMA2 | 270 | 1 | October to November 2023 |
| Item order analysis | GPT-3.5 | 18 | 12–15 | April to May 2023 |
| False belief perturbations | Human | 757 | 1 | November 2023 |
| | GPT-4 | 225 | 1 | October to November 2023 |
| | GPT-3.5 | 225 | 1 | October to November 2023 |
| | LLaMA2 | 225 | 1 | October to November 2023 |

N, human participants; n, independent LLM observations. Details of data collection for each model at each stage of the study are shown, including N (human participants)/n (independent observations of LLM responses), number of items administered to each individual observation (ranges where multiple tests were administered) and dates of data collection. Information is the same for LlaMA2-70B, LlaMA2-13B and LlaMA2-7B. Analysis of the data in the item order analysis and false belief perturbations is reported in Supplementary Information sections 3 and 4.

real-time human–machine interactions[54,55]. Failure of commitment by GPT models, for example, may lead to negative affect in human conversational partners. However, it may also foster curiosity[40]. Understanding how GPTs' performance on mentalistic inferences (or their absences) influences human social cognition in dynamically unfolding social interactions is an open challenge for future work.

The LLM landscape is fast-moving. Our findings highlight the importance of systematic testing and proper validation in human samples as a necessary foundation. As artificial intelligence (AI) continues to evolve, it also becomes increasingly important to heed calls for open science and open access to these models[32]. Direct access to the parameters, data and documentation used to construct models can allow for targeted probing and experimentation into the key parameters affecting social reasoning, informed by and building on comparisons with human data. As such, open models can not only serve to accelerate the development of future AI technologies but also serve as models of human cognition.

## Methods
### Ethical compliance
The research was approved by the local ethical committee (ASL 3 Genovese; protocol no. 192REG2015) and was carried out in accordance with the principles of the revised Helsinki Declaration.

### Experimental model details
We tested two versions of OpenAI's GPT: version 3.5, which was the default model at the time of testing, and version 4, which was the state-of-the-art model with enhanced reasoning, creativity and comprehension relative to previous models (https://chat.openai.com/). Each test was delivered in a separate chat: GPT is capable of learning within a chat session, as it can remember both its own and the user's previous messages to adapt its responses accordingly, but it does not retain this memory across new chats. As such, each new iteration of a test may be considered a blank slate with a new naive participant. The dates of data collection for the different stages are reported in Table 1.

Three LLaMA2-Chat models were tested. These models were trained on sets of different sizes: 70, 13 and 7 billion tokens. All LLaMA2-Chat responses were collected using set parameters with the prompt, 'You are a helpful AI assistant', a temperature of 0.7, the maximum number of new tokens set at 512, a repetition penalty of 1.1, and a Top P of 0.9. Langchain's conversation chain was used to create a memory context within individual chat sessions. Responses from all LLaMA2-Chat models were found to include a number of non-codable responses (for example, repeating the question without answering it), and these were regenerated individually and included with the full response set. For the 70B model, these non-responses were rare, but for the 13B and 7B models they were common enough to cause concern about the quality of these data. As such, only the responses of the 70B model are reported in the main manuscript and a comparison of this model against the smaller two is reported in Supplementary Information section 1. Details and dates of data collection are reported in Table 1.

For each test, we collected 15 sessions for each LLM. A session involved delivering all items of a single test within the same chat window. GPT-4 was subject to a 25-message limit per 3 h; to minimize interference, a single experimenter delivered all tests for GPT-4, while four other experimenters shared the duty of collecting responses from GPT-3.5.

Human participants were recruited online through the Prolific platform and the study was hosted on SoSci. We recruited native English speakers between the ages of 18 and 70 years with no history of psychiatric conditions and no history of dyslexia in particular. Further demographic data were not collected. We aimed to collect around 50 participants per test (theory of mind battery) or item (belief likelihood test, false belief perturbations). Thirteen participants who appeared to have generated their answers using LLMs or whose responses did not answer the questions were excluded. The final human sample was N = 1,907 (Table 1). All participants provided informed consent through the online survey and received monetary compensation in return for their participation at a rate of GBP£12 h⁻¹.

### Theory of mind battery
We selected a series of tests typically used in evaluating theory of mind capacity in human participants.

**False belief.** False belief assess the ability to infer that another person possesses knowledge that differs from the participant's own (true) knowledge of the world. These tests consist of test items that follow a particular structure: character A and character B are together, character A deposits an item inside a hidden location (for example, a box), character A leaves, character B moves the item to a second hidden location (for example, a cupboard) and then character A returns. The question asked to the participant is: when character A returns, will they look for the item in the new location (where it truly is, matching the participant's true belief) or the old location (where it was, matching character A's false belief)?

In addition to the false belief condition, the test also uses a true belief control condition, where rather than move the item that character A hid, character B moves a different item to a new location. This is important for interpreting failures of false belief attribution as they ensure that any failures are not due to a recency effect (referring to the last location reported) but instead reflect an accurate belief tracking.

We adapted four false/true belief scenarios from the sandbox task used by Bernstein[33] and generated three novel items, each with false and

true belief versions. These novel items followed the same structure as the original published items but with different details such as names, locations or objects to control for familiarity with the text of published items. Two story lists (false belief A, false belief B) were generated for this test such that each story only appeared once within a testing session and alternated between false and true belief depending on the session. In addition to the standard false/true belief scenarios, two additional catch stories were tested that involved minor alterations to the story structure. The results of these items are not reported here as they go beyond the goals of the current study.

**Irony.** Comprehending an ironic remark requires inferring the true meaning of an utterance (typically the opposite of what is said) and detecting the speaker's mocking attitude, and this has been raised as a key challenge for AI and LLMs[19].

Irony comprehension items were adapted from an eye-tracking study[34] in which participants read vignettes where a character made an ironic or non-ironic statement. Twelve items were taken from these stimuli that in the original study were used as comprehension checks. Items were abbreviated to end following the ironic or non-ironic utterance.

Two story lists were generated for this test (irony A, irony B) such that each story only appeared once within a testing session and alternated between ironic and non-ironic depending on the session. Responses were coded as 1 (correct) or 0 (incorrect). During coding, we noted some inconsistencies in the formulation of both GPT models' responses where in response to the question of whether the speaker believed what they had said, they might respond with, 'Yes, they did not believe that…'. Such internally contradictory responses, where the models responded with a 'yes' or 'no' that was incompatible with the follow-up explanation, were coded on the basis of whether or not the explanation showed appreciation of the irony—the linguistic failures of these models in generating a coherent answer are not of direct interest to the current study as these failures (1) were rare and (2) did not render the responses incomprehensible.

**Faux pas.** The faux pas test[13] presents a context in which one character makes an utterance that is unintentionally offensive to the listener because the speaker does not know or does not remember some key piece of information.

Following the presentation of the scenario, we presented four questions:

1. 'In the story did someone say something that they should not have said?' [The correct answer is always 'yes']
2. 'What did they say that they should not have said?' [Correct answer changes for each item]
3. A comprehension question to test understanding of story events [Question changes for every item]
4. A question to test awareness of the speaker's false belief phrased as, 'Did [the speaker] know that [what they said was inappropriate]?' [Question changes for every item. The correct answer is always 'no']

These questions were asked at the same time as the story was presented. Under the original coding criteria, participants must answer all four questions correctly for their answer to be considered correct. However, in the current study we were interested primarily in the response to the final question testing whether the responder understood the speaker's mental state. When examining the human data, we noticed that several participants responded incorrectly to the first item owing to an apparent unwillingness to attribute blame (for example 'No, he didn't say anything wrong because he forgot'). To focus on the key aspect of faux pas understanding that was relevant to the current study, we restricted our coding to only the last question (1 (correct if the answer was no) or 0 (for anything else); see Supplementary Information

section 5 for an alternative coding that follows the original criteria, as well as a recoding where we coded as correct responses where the correct answer was mentioned as a possible explanation but was not explicitly endorsed).

As well as the 10 original items used in Baron-Cohen et al.[13], we generated five novel items for this test that followed the same structure and logic as the original items, resulting in 15 items overall.

**Hinting task.** The hinting task[14] assesses the understanding of indirect speech requests through the presentation of ten vignettes depicting everyday social interactions that are presented sequentially. Each vignette ends with a remark that can be interpreted as a hint.

A correct response identifies both the intended meaning of the remark and the action that it is attempting to elicit. In the original test, if the participant failed to answer the question fully the first time, they were prompted with additional questioning[14,56]. In our adapted implementation, we removed this additional questioning and coded responses as a binary (1 (correct) or 0 (incorrect)) using the evaluation criteria listed in Gil et al.[56]. Note that this coding offers more conservative estimates of hint comprehension than in previous studies.

In addition to 10 original items sourced from Corcoran[14], we generated a further 6 novel hinting test items, resulting in 16 items overall.

**Strange stories.** The strange stories[15,16] offer a means of testing more advanced mentalizing abilities such as reasoning about misdirection, manipulation, lying and misunderstanding, as well as second- or higher-order mental states (for example, A knows that B believes $X$…). The advanced abilities that these stories measure make them suitable for testing higher-functioning children and adults. In this test, participants are presented with a short vignette and are asked to explain why a character says or does something that is not literally true.

Each question comes with a specific set of coding criteria and responses can be awarded 0, 1 or 2 points depending on how fully it explains the utterance and whether or not it explains it in mentalistic terms[16]. See Supplementary Information section 6 for a description of the frequency of partial successes.

In addition to the 8 original mental stories, we generated 4 novel items, resulting in 12 items overall. The maximum number of points possible was 24, and individual session scores were converted to a proportional score for analysis.

**Testing protocol.** For the theory of mind battery, the order of items was set for each test, with original items delivered first and novel items delivered last. Each item was preceded by a preamble that remained consistent across all tests. This was then followed by the story description and the relevant question(s). After each item was delivered, the model would respond and then the session advanced to the next item.

For GPT models, items were delivered using the chat web interface. For LLaMA2-Chat models, delivery of items was automated through a custom script. For humans, items were presented with free text response boxes on separate pages of a survey so that participants could write out their responses to each question (with a minimum character count of 2).

### Faux pas likelihood test

To test alternative hypotheses of why the tested models performed poorly at the faux pas test, we ran a follow-up study replicating just the faux pas test. This replication followed the same procedure as the main study with one major difference.

The original wording of the question was phrased as a straightforward yes/no question that tested the subject's awareness of a speaker's false belief (for example, 'Did Richard remember James had given him the toy aeroplane for his birthday?'). To test whether the low scores on this question were due to the models' refusing to commit to a single explanation in the face of ambiguity, we reworded this to ask in terms

of likelihood: 'Is it more likely that Richard remembered or did not remember that James had given him the toy aeroplane for his birthday?'

Another difference from the original study was that we included a follow-up prompt in the rare cases where the model failed to provide clear reasoning on an incorrect response. The coding criteria for this follow-up were in line with coding schemes used in other studies with a prompt system[14], where an unprompted correct answer was given 2 points, a correct answer following a prompt was given 1 point and incorrect answers following a prompt were given 0 points. These points were then rescaled to a proportional score to allow comparison against the original wording.

During coding by the human experimenters, a qualitative description of different subtypes of response (beyond 0–1–2 points) emerged, particularly noting recurring patterns in responses that were marked as successes. This exploratory qualitative breakdown is reported along with further detail on the prompting protocol in Supplementary Information section 7.

### Belief likelihood test

To manipulate the likelihood that the speaker knew or did not know, we developed a new set of variants of the faux pas likelihood test. For each test item, all newly generated for this control study, we created three variants: a faux pas variant, a neutral variant and a knowledge-implied variant. In the faux pas variant, the utterance suggested that the speaker did not know the context. In the neutral variant, the utterance suggested neither that they knew nor did not know. In the knowledge-implied variant, the utterance suggested that the speaker knew (for the full text of all items, see Supplementary Appendix 2). For each variant, the core story remained unchanged, for example:

```
Michael was a very awkward child when he was at
high school. He struggled with making friends
and spent his time alone writing poetry. However,
after he left he became a lot more confident and
sociable. At his ten-year high school reunion he
met Amanda, who had been in his English class. Over
drinks, she said to him,
```

followed by the utterance, which varied across conditions:
Faux Pas:

```
'I don't know if you remember this guy from school.
He was in my English class. He wrote poetry and he
was super awkward. I hope he isn't here tonight.'
```

Neutral:

```
'Do you know where the bar is?'
```

Knowledge implied:

```
'Do you still write poetry?'
```

The belief likelihood test was administered in the same way as with previous tests with the exception that responses were kept independent so that there was no risk of responses being influenced by other variants. For ChatGPT models, this involved delivering each item within a separate chat session for 15 repetitions of each item. For LLaMA2-70B, this involved removing the Langchain conversation chain allowing for within-session memory context. Human participants were recruited separately to answer a single test item, with at least 50 responses collected for each item (total $N = 900$). All other details of the protocol were the same.

### Quantification and statistical analysis
**Response coding.** After each session in the theory of mind battery and faux pas likelihood test, the responses were collated and coded by five

human experimenters according to the pre-defined coding criteria for each test. Each experimenter was responsible for coding 100% of sessions for one test and 20% of sessions for another. Inter-coder per cent agreement was calculated on the 20% of shared sessions, and items where coders showed disagreement were evaluated by all raters and recoded. The data available on the OSF are the results of this recoding. Experimenters also flagged individual responses for group evaluation if they were unclear or unusual cases, as and when they arose. Inter-rater agreement was computed by calculating the item-wise agreement between coders as 1 or 0 and using this to calculate a percentage score. Initial agreement across all double-coded items was over 95%. The lowest agreement was for the human and GPT-3.5 responses of strange stories, but even here agreement was over 88%. Committee evaluation by the group of experimenters resolved all remaining ambiguities.

For the belief likelihood test, responses were coded according to whether they endorsed the 'knew' explanation or 'didn't know' explanation, or whether they did not endorse either as more likely than the other. Outcomes 'knew', 'unsure' and 'didn't know' were assigned a numerical coding of +1, 0 and −1, respectively. GPT models adhered closely to the framing of the question in their answer, but humans were more variable and sometimes provided ambiguous responses (for example, 'yes', 'more likely' and 'not really') or did not answer the question at all ('It doesn't matter' and 'She didn't care'). These responses were rare, constituting only ~2.5% of responses and were coded as endorsing the 'knew' explanation if they were affirmative ('yes') and the 'didn't know' explanation if they were negative.

### Statistical analysis
**Comparing LLMs against human performance.** Scores for individual responses were scaled and averaged to obtain a proportional score for each test session in order to create a performance metric that could be compared directly across different theory of mind tests. Our goal was to compare LLMs' performance across different tests against human performance to see how these models performed on theory of mind tests relative to humans. For each test, we compared the performance of each of the three LLMs against human performance using a set of Holm-corrected two-way Wilcoxon tests. Effect sizes for Wilcoxon tests were calculated by dividing the test statistic $Z$ by the square root of the total sample size, and 95% CIs of the effect size were bootstrapped over 1,000 iterations. All non-significant results were further examined using corresponding Bayesian tests represented as a Bayes factor ($BF_{10}$) under continuous prior distribution (Cauchy prior width $r = 0.707$). Bayes factors were computed in JASP 0.18.3 with a random seed value of 1. The results of the false belief test were not subjected to inferential statistics owing to the ceiling performance and lack of variance across models.

**Novel items.** For each publicly available test (all tests except for irony), we generated novel items that followed the same logic as the original text but with different details and text to control for low-level familiarity with the scenarios through inclusion in the LLM training sets. For each of these tests, we compared the performance of all LLMs on these novel items against the validated test items using Holm-corrected two-way Wilcoxon tests. Non-significant results were followed up with corresponding Bayesian tests in JASP. Significantly poorer performance on novel items than original items would indicate a strong likelihood that the good performance of a language model can be attributed to inclusion of these texts in the training set. Note that, while the open-ended format of more complex tasks like hinting and strange stories makes this a convincing control for these tests, they are of limited strength for tasks like false belief and faux pas that use a regular internal structure that make heuristics or 'Clever Hans' solutions possible[27,36].

**Belief likelihood test.** We calculated the count frequency of the different response types ('didn't know', 'unsure' and 'knew') for each variant and each model. Then, for each model we conducted two chi-square

tests that compared the distribution of these categorical responses to the faux pas variant against the neutral, and to the neutral variant against the knowledge implied. A Holm correction was applied to the eight chi-square tests to account for multiple comparisons. The non-significant result was further examined with a Bayesian contingency table in JASP.

## Reporting summary

Further information on research design is available in the Nature Portfolio Reporting Summary linked to this article.

## Data availability

All resources are available on a repository stored on the Open Science Framework (OSF) under a Creative Commons Attribution Non-Commercial 4.0 International (CC-BY-NC) license at https://osf.io/fwj6v. This repository contains all test items, data and code reported in this study. Test items and data are available in an Excel file that includes the text of every item delivered in each test, the full text responses to each item and the code assigned to each response. This file is available at https://osf.io/dbn92 Source data are provided with this paper.

## Code availability

The code used for all analysis in the main manuscript and Supplementary Information is included as a Markdown file at https://osf.io/fwj6v. The data used by the analysis files are available as a number of CSV files under 'scored_data/' in the repository, and all materials necessary for replicating the analysis can be downloaded as a single .zip file within the main repository titled 'Full R Project Code.zip' at https://osf.io/j3vhq.

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

## Acknowledgements

This work is supported by the European Commission through Project ASTOUND (101071191—HORIZON-EIC-2021-PATHFINDERCHALLENGES-01 to A.R., G.M., C.B. and S.P.). J.W.A.S. was supported by a Humboldt Research Fellowship for Experienced Researchers provided by the Alexander von Humboldt Foundation. The funders had no role in study design, data collection and analysis, decision to publish or preparation of the manuscript.

## Author contributions

J.W.A.S., A.R., G.M., M.S.A.G. and C.B. conceived the study. J.W.A.S., D.A., G.B., O.P. and E.S. designed the tasks and performed the experiments including data collection with humans and GPT models, response coding and curation of the dataset. S.G., K.S. and G.M. collected data from LLaMA2-Chat models. J.W.A.S. performed the analyses and wrote the manuscript with input from C.B., S.P. and M.S.A.G. All authors contributed to the interpretation and editing of the manuscript. C.B. supervised the work. A.R., G.M., S.P. and C.B. acquired the funding. D.A., G.B., O.P. and E.S. contributed equally to the work.

## Funding

## Competing interests

The authors declare no competing interests.

## Additional information

**Correspondence and requests for materials** should be addressed to James W. A. Strachan or Cristina Becchio.

# Reporting Summary

## Statistics

For all statistical analyses, confirm that the following items are present in the figure legend, table legend, main text, or Methods section.

| n/a | Confirmed | |
|---|---|---|
| ☐ | ☒ | The exact sample size (*n*) for each experimental group/condition, given as a discrete number and unit of measurement |
| ☐ | ☒ | A statement on whether measurements were taken from distinct samples or whether the same sample was measured repeatedly |
| ☐ | ☒ | The statistical test(s) used AND whether they are one- or two-sided *Only common tests should be described solely by name; describe more complex techniques in the Methods section.* |
| ☐ | ☒ | A description of all covariates tested |
| ☐ | ☒ | A description of any assumptions or corrections, such as tests of normality and adjustment for multiple comparisons |
| ☐ | ☒ | A full description of the statistical parameters including central tendency (e.g. means) or other basic estimates (e.g. regression coefficient) AND variation (e.g. standard deviation) or associated estimates of uncertainty (e.g. confidence intervals) |
| ☐ | ☒ | For null hypothesis testing, the test statistic (e.g. *F*, *t*, *r*) with confidence intervals, effect sizes, degrees of freedom and *P* value noted *Give P values as exact values whenever suitable.* |
| ☐ | ☒ | For Bayesian analysis, information on the choice of priors and Markov chain Monte Carlo settings |
| ☒ | ☐ | For hierarchical and complex designs, identification of the appropriate level for tests and full reporting of outcomes |
| ☐ | ☒ | Estimates of effect sizes (e.g. Cohen's *d*, Pearson's *r*), indicating how they were calculated |

*Our web collection on statistics for biologists contains articles on many of the points above.*

## Software and code

Policy information about availability of computer code

| Data collection | Human behavioural data was collected in online experiments using the Prolific platform directing to a survey hosted on the SoSci platform. Data from GPT models was collected through the chat web interface at http://chat.openai.com. A custom script automated the delivery of questions and collection of data for LLaMA2-Chat models, which are available from https://www.llama2.ai/ |
|---|---|
| Data analysis | We used R for data analysis and for creating the figures<br>R version 4.1.2<br>RStudio 2024.04.0-daily+368 "Chocolate Cosmos" Daily (605bbb38ebb4f8565e361122f6d8be3486d288e9, 2024-02-01) for Ubuntu Jammy<br>Mozilla/5.0 (X11; Linux x86_64) AppleWebKit/537.36 (KHTML, like Gecko) rstudio/2024.04.0-daily+368 Chrome/120.0.6099.56<br>Electron/28.0.0 Safari/537.36<br>The code used for data analysis is available as a stand-alone RMarkdown project from: https://osf.io/j3vhq<br>This code uses the following R packages:<br>DescTools_0.99.50<br>flextable_0.9.4<br>kableExtra_1.3.4<br>rstatix_0.7.2<br>cowplot_1.1.2<br>ggdist_3.3.1<br>ggpubr_0.6.0<br>ggplot2_3.4.4<br>purrr_1.0.2<br>Hmisc_5.1-1 |

```
tidyr_1.3.0
dplyr_1.1.4
ggtext_0.1.2
Null results reported in the main manuscript were subjected to follow-up corresponding Bayesian analyses to compute Bayes Factors (BF10).
This analysis was done using JASP v0.18.3 (JASP Team, 2024)
```

For manuscripts utilizing custom algorithms or software that are central to the research but not yet described in published literature, software must be made available to editors and reviewers. We strongly encourage code deposition in a community repository (e.g. GitHub). See the Nature Portfolio guidelines for submitting code & software for further information.

## Data

Policy information about availability of data

All manuscripts must include a data availability statement. This statement should provide the following information, where applicable:
- Accession codes, unique identifiers, or web links for publicly available datasets
- A description of any restrictions on data availability
- For clinical datasets or third party data, please ensure that the statement adheres to our policy

All data reported in the current study can be found in an OSF repository under a Creative Commons Attribution Non-Commercial 4.0 International license (CC-BY-NC). The repository can be accessed at the following URL: https://osf.io/fwj6v/
The full text of question items, the full text of responses from GPT models, LLaMA2 models, and human participants, and the scores assigned to each response can be downloaded as a single file from the following URL: https://osf.io/dbn92
Data files with scores alone, which can be used to recreate the analysis, are stored in the OSF repository in the folder scored_data/

## Research involving human participants, their data, or biological material

Policy information about studies with human participants or human data. See also policy information about sex, gender (identity/presentation), and sexual orientation and race, ethnicity and racism.

| | |
|---|---|
| Reporting on sex and gender | Data on sex and gender were not collected. |
| Reporting on race, ethnicity, or other socially relevant groupings | Data on race and ethnicity were not collected. |
| Population characteristics | We recruited native English speakers between the ages of 18 and 70 with no history of psychiatric conditions and no history of dyslexia. Further demographic data were not collected. |
| Recruitment | Participants were recruited through the online platform Prolific and were compensated at an adjusted rate of GBP£12/hr (between £2-£6). To our knowledge, there were no significant sources of self-selection bias that would be likely to impact the study findings as a result of this recruitment procedure. |
| Ethics oversight | The research was approved by the local ethics committee (ASL 3 Genovese) and was carried out in accordance with the principles of the revised Helsinki Declaration. |

Note that full information on the approval of the study protocol must also be provided in the manuscript.

## Field-specific reporting

Please select the one below that is the best fit for your research. If you are not sure, read the appropriate sections before making your selection.

☐ Life sciences ☒ Behavioural & social sciences ☐ Ecological, evolutionary & environmental sciences

For a reference copy of the document with all sections, see nature.com/documents/nr-reporting-summary-flat.pdf

## Life sciences study design

All studies must disclose on these points even when the disclosure is negative.

| | |
|---|---|
| Sample size | *Describe how sample size was determined, detailing any statistical methods used to predetermine sample size OR if no sample-size calculation was performed, describe how sample sizes were chosen and provide a rationale for why these sample sizes are sufficient.* |
| Data exclusions | *Describe any data exclusions. If no data were excluded from the analyses, state so OR if data were excluded, describe the exclusions and the rationale behind them, indicating whether exclusion criteria were pre-established.* |
| Replication | *Describe the measures taken to verify the reproducibility of the experimental findings. If all attempts at replication were successful, confirm this OR if there are any findings that were not replicated or cannot be reproduced, note this and describe why.* |
| Randomization | *Describe how samples/organisms/participants were allocated into experimental groups. If allocation was not random, describe how covariates were controlled OR if this is not relevant to your study, explain why.* |

| Blinding | *Describe whether the investigators were blinded to group allocation during data collection and/or analysis. If blinding was not possible, describe why OR explain why blinding was not relevant to your study.* |

# Behavioural & social sciences study design

All studies must disclose on these points even when the disclosure is negative.

| Study description | The data consist of full-text responses to questions on a set of Theory of Mind tests. Data reported in the manuscript are quantitative numeric scores assigned to each text response according to published coding criteria, with any deviations from validated procedures clearly highlighted in the Methods of the main manuscript. The design is a between-samples comparison of three Large Language Models (LLMs) against a baseline sample of human respondents. |

| Research sample | LLMs: GPT-4, GPT-3.5, LLaMA2-70B (and other LLaMA2 models reported in Supplementary Information): 15 administrations of each test (sessions); Humans: target N of 50 unique participants for each test, total N=1907 (between-subjects). No additional demographic information was collected, but only native English speakers between 18 and 70 with no history of dyslexia or psychiatric conditions were recruited in order to ensure that they could complete the task and read the stories. We did not specify particular demographics or collect this data because the main comparison of interest was human vs. LLM performance and we had no reason to build a priori hypotheses about specific demographics. Recruitment was not restricted to any country and was not restricted to reflect a representative distribution of UK or US census data. |

| Sampling strategy | Convenience sample through the Prolific platform. Participants were paid GBP£12/hr for participation (between £2-£6, depending on the test). The sample size was set based on the control adult sample size of White et al. (2009), which recruited 40 neurotypical adults for an update and validation of the Strange Stories task (which, as the most difficult task of the battery, we considered the most likely to show variability). To account for any data quality issues posed by online data collection, we rounded up the target sample size to N=50 per test. |

| Data collection | For each test we collected 15 sessions for each LLM and ~50 human subjects through Prolific. GPT models were tested through the OpenAI ChatGPT web interface, and a session involved delivering all items of a single test within the same chat window. LLaMA models were tested using Langchain using set parameters with the prompt, "You are a helpful AI assistant", a temperature of 0.7, the maximum number of new tokens set at 512, a repetition penalty of 1.1, and a top P of 0.9. For humans, all items were presented sequentially through an online survey built and hosted through the SoSci platform. Experimenters were not blinded to the experimental conditions as there was no reciprocal interaction with the participants. In the case of the Faux Pas Likelihood test, which included the experimenter delivering a follow-up prompt in the case of unclear reasoning on an incorrect answer from GPT models, criteria for deciding to deliver the follow-up were set a priori and evaluated afterwards by other experimenters to check that the prompt had been valid. |

| Timing | The GPT data on the full battery reported in the main manuscript and in the supplementary material were collected between 3 April and 18 April 2023. The follow-up data using an adapted version of the Faux Pas test were collected between 28 April and 4 May 2023. The follow-up data with GPT-3.5 using a randomised presentation order on the Irony, Strange Stories, and Faux Pas tests were collected between 24 April and 18 May 2023. Three LLaMA2-Chat models were tested between October and November 2023. Variant testing of the False Belief and Faux Pas tests (Belief Likelihoood test) for GPT models occurred between 25 October and 3 November 2023. |

| Data exclusions | Thirteen (13) human subjects were excluded from final analysis following initial examination of the data. Theory of Mind Battery: two (2) subjects who used GPT or another LLM to answer the questions and one (1) subject who just responded 'Yes' to every question; Belief Likelihood Test: seven (7) participants who were believed to use GPT or another LLM to generate their responses; False Belief Perturbations: three (3) participants who were believed to use GPT or another LLM to generate their responses. |

| Non-participation | No participants dropped out or declined participation. |

| Randomization | Participants were not assigned to experimental groups, but volunteered to complete one of the five Theory of Mind tests. This was a random opportunity sample, and individuals who had participated in one test were excluded from participating again. |

# Ecological, evolutionary & environmental sciences study design

All studies must disclose on these points even when the disclosure is negative.

| Study description | *Briefly describe the study. For quantitative data include treatment factors and interactions, design structure (e.g. factorial, nested, hierarchical), nature and number of experimental units and replicates.* |

| Research sample | *Describe the research sample (e.g. a group of tagged Passer domesticus, all Stenocereus thurberi within Organ Pipe Cactus National Monument), and provide a rationale for the sample choice. When relevant, describe the organism taxa, source, sex, age range and any manipulations. State what population the sample is meant to represent when applicable. For studies involving existing datasets, describe the data and its source.* |

| Sampling strategy | *Note the sampling procedure. Describe the statistical methods that were used to predetermine sample size OR if no sample-size calculation was performed, describe how sample sizes were chosen and provide a rationale for why these sample sizes are sufficient.* |

| Data collection | *Describe the data collection procedure, including who recorded the data and how.* |

| Timing and spatial scale | *Indicate the start and stop dates of data collection, noting the frequency and periodicity of sampling and providing a rationale for these choices. If there is a gap between collection periods, state the dates for each sample cohort. Specify the spatial scale from which the data are taken* |
|---|---|
| Data exclusions | *If no data were excluded from the analyses, state so OR if data were excluded, describe the exclusions and the rationale behind them, indicating whether exclusion criteria were pre-established.* |
| Reproducibility | *Describe the measures taken to verify the reproducibility of experimental findings. For each experiment, note whether any attempts to repeat the experiment failed OR state that all attempts to repeat the experiment were successful.* |
| Randomization | *Describe how samples/organisms/participants were allocated into groups. If allocation was not random, describe how covariates were controlled. If this is not relevant to your study, explain why.* |
| Blinding | *Describe the extent of blinding used during data acquisition and analysis. If blinding was not possible, describe why OR explain why blinding was not relevant to your study.* |

Did the study involve field work? ☐ Yes ☐ No

## Field work, collection and transport

| Field conditions | *Describe the study conditions for field work, providing relevant parameters (e.g. temperature, rainfall).* |
|---|---|
| Location | *State the location of the sampling or experiment, providing relevant parameters (e.g. latitude and longitude, elevation, water depth).* |
| Access & import/export | *Describe the efforts you have made to access habitats and to collect and import/export your samples in a responsible manner and in compliance with local, national and international laws, noting any permits that were obtained (give the name of the issuing authority, the date of issue, and any identifying information).* |
| Disturbance | *Describe any disturbance caused by the study and how it was minimized.* |

# Reporting for specific materials, systems and methods

We require information from authors about some types of materials, experimental systems and methods used in many studies. Here, indicate whether each material, system or method listed is relevant to your study. If you are not sure if a list item applies to your research, read the appropriate section before selecting a response.

| Materials & experimental systems | | | Methods | | |
|---|---|---|---|---|---|
| n/a | Involved in the study | | n/a | Involved in the study | |
| ☒ | ☐ Antibodies | | ☒ | ☐ ChIP-seq | |
| ☒ | ☐ Eukaryotic cell lines | | ☒ | ☐ Flow cytometry | |
| ☒ | ☐ Palaeontology and archaeology | | ☒ | ☐ MRI-based neuroimaging | |
| ☒ | ☐ Animals and other organisms | | | | |
| ☒ | ☐ Clinical data | | | | |
| ☒ | ☐ Dual use research of concern | | | | |
| ☒ | ☐ Plants | | | | |

## Antibodies

| Antibodies used | *Describe all antibodies used in the study; as applicable, provide supplier name, catalog number, clone name, and lot number.* |
|---|---|
| Validation | *Describe the validation of each primary antibody for the species and application, noting any validation statements on the manufacturer's website, relevant citations, antibody profiles in online databases, or data provided in the manuscript.* |

## Eukaryotic cell lines

Policy information about cell lines and Sex and Gender in Research

| Cell line source(s) | *State the source of each cell line used and the sex of all primary cell lines and cells derived from human participants or vertebrate models.* |
|---|---|
| Authentication | *Describe the authentication procedures for each cell line used OR declare that none of the cell lines used were authenticated.* |

| Mycoplasma contamination | *Confirm that all cell lines tested negative for mycoplasma contamination OR describe the results of the testing for mycoplasma contamination OR declare that the cell lines were not tested for mycoplasma contamination.* |
|---|---|
| Commonly misidentified lines<br>(See ICLAC register) | *Name any commonly misidentified cell lines used in the study and provide a rationale for their use.* |

# Palaeontology and Archaeology

| Specimen provenance | *Provide provenance information for specimens and describe permits that were obtained for the work (including the name of the issuing authority, the date of issue, and any identifying information). Permits should encompass collection and, where applicable, export.* |
|---|---|
| Specimen deposition | *Indicate where the specimens have been deposited to permit free access by other researchers.* |
| Dating methods | *If new dates are provided, describe how they were obtained (e.g. collection, storage, sample pretreatment and measurement), where they were obtained (i.e. lab name), the calibration program and the protocol for quality assurance OR state that no new dates are provided.* |

☐ Tick this box to confirm that the raw and calibrated dates are available in the paper or in Supplementary Information.

| Ethics oversight | *Identify the organization(s) that approved or provided guidance on the study protocol, OR state that no ethical approval or guidance was required and explain why not.* |
|---|---|

Note that full information on the approval of the study protocol must also be provided in the manuscript.

# Animals and other research organisms

Policy information about studies involving animals; ARRIVE guidelines recommended for reporting animal research, and Sex and Gender in Research

| Laboratory animals | *For laboratory animals, report species, strain and age OR state that the study did not involve laboratory animals.* |
|---|---|
| Wild animals | *Provide details on animals observed in or captured in the field; report species and age where possible. Describe how animals were caught and transported and what happened to captive animals after the study (if killed, explain why and describe method; if released, say where and when) OR state that the study did not involve wild animals.* |
| Reporting on sex | *Indicate if findings apply to only one sex; describe whether sex was considered in study design, methods used for assigning sex. Provide data disaggregated for sex where this information has been collected in the source data as appropriate; provide overall numbers in this Reporting Summary. Please state if this information has not been collected. Report sex-based analyses where performed, justify reasons for lack of sex-based analysis.* |
| Field-collected samples | *For laboratory work with field-collected samples, describe all relevant parameters such as housing, maintenance, temperature, photoperiod and end-of-experiment protocol OR state that the study did not involve samples collected from the field.* |
| Ethics oversight | *Identify the organization(s) that approved or provided guidance on the study protocol, OR state that no ethical approval or guidance was required and explain why not.* |

Note that full information on the approval of the study protocol must also be provided in the manuscript.

# Clinical data

Policy information about clinical studies
All manuscripts should comply with the ICMJE guidelines for publication of clinical research and a completed CONSORT checklist must be included with all submissions.

| Clinical trial registration | *Provide the trial registration number from ClinicalTrials.gov or an equivalent agency.* |
|---|---|
| Study protocol | *Note where the full trial protocol can be accessed OR if not available, explain why.* |
| Data collection | *Describe the settings and locales of data collection, noting the time periods of recruitment and data collection.* |
| Outcomes | *Describe how you pre-defined primary and secondary outcome measures and how you assessed these measures.* |

# Dual use research of concern

Policy information about dual use research of concern

## Hazards

Could the accidental, deliberate or reckless misuse of agents or technologies generated in the work, or the application of information presented in the manuscript, pose a threat to:

No | Yes

☐ | ☐ Public health

☐ | ☐ National security

☐ | ☐ Crops and/or livestock

☐ | ☐ Ecosystems

☐ | ☐ Any other significant area

## Experiments of concern

Does the work involve any of these experiments of concern:

No | Yes

☐ | ☐ Demonstrate how to render a vaccine ineffective

☐ | ☐ Confer resistance to therapeutically useful antibiotics or antiviral agents

☐ | ☐ Enhance the virulence of a pathogen or render a nonpathogen virulent

☐ | ☐ Increase transmissibility of a pathogen

☐ | ☐ Alter the host range of a pathogen

☐ | ☐ Enable evasion of diagnostic/detection modalities

☐ | ☐ Enable the weaponization of a biological agent or toxin

☐ | ☐ Any other potentially harmful combination of experiments and agents

# Plants

| | |
|---|---|
| Seed stocks | *Report on the source of all seed stocks or other plant material used. If applicable, state the seed stock centre and catalogue number. If plant specimens were collected from the field, describe the collection location, date and sampling procedures.* |
| Novel plant genotypes | *Describe the methods by which all novel plant genotypes were produced. This includes those generated by transgenic approaches, gene editing, chemical/radiation-based mutagenesis and hybridization. For transgenic lines, describe the transformation method, the number of independent lines analyzed and the generation upon which experiments were performed. For gene-edited lines, describe the editor used, the endogenous sequence targeted for editing, the targeting guide RNA sequence (if applicable) and how the editor was applied.* |
| Authentication | *Describe any authentication procedures for each seed stock used or novel genotype generated. Describe any experiments used to assess the effect of a mutation and, where applicable, how potential secondary effects (e.g. second site T-DNA insertions, mosiacism, off-target gene editing) were examined.* |

# ChIP-seq

## Data deposition

☐ Confirm that both raw and final processed data have been deposited in a public database such as GEO.

☐ Confirm that you have deposited or provided access to graph files (e.g. BED files) for the called peaks.

| | |
|---|---|
| Data access links<br>*May remain private before publication.* | *For "Initial submission" or "Revised version" documents, provide reviewer access links.  For your "Final submission" document, provide a link to the deposited data.* |
| Files in database submission | *Provide a list of all files available in the database submission.* |
| Genome browser session<br>(e.g. UCSC) | *Provide a link to an anonymized genome browser session for "Initial submission" and "Revised version" documents only, to enable peer review.  Write "no longer applicable" for "Final submission" documents.* |

## Methodology

| | |
|---|---|
| Replicates | *Describe the experimental replicates, specifying number, type and replicate agreement.* |
| Sequencing depth | *Describe the sequencing depth for each experiment, providing the total number of reads, uniquely mapped reads, length of reads and whether they were paired- or single-end.* |
| Antibodies | *Describe the antibodies used for the ChIP-seq experiments; as applicable, provide supplier name, catalog number, clone name, and lot number.* |

| Peak calling parameters | *Specify the command line program and parameters used for read mapping and peak calling, including the ChIP, control and index files used.* |
|---|---|
| Data quality | *Describe the methods used to ensure data quality in full detail, including how many peaks are at FDR 5% and above 5-fold enrichment.* |
| Software | *Describe the software used to collect and analyze the ChIP-seq data. For custom code that has been deposited into a community repository, provide accession details.* |

# Flow Cytometry

## Plots

Confirm that:

- ☐ The axis labels state the marker and fluorochrome used (e.g. CD4-FITC).
- ☐ The axis scales are clearly visible. Include numbers along axes only for bottom left plot of group (a 'group' is an analysis of identical markers).
- ☐ All plots are contour plots with outliers or pseudocolor plots.
- ☐ A numerical value for number of cells or percentage (with statistics) is provided.

## Methodology

| Sample preparation | *Describe the sample preparation, detailing the biological source of the cells and any tissue processing steps used.* |
|---|---|
| Instrument | *Identify the instrument used for data collection, specifying make and model number.* |
| Software | *Describe the software used to collect and analyze the flow cytometry data. For custom code that has been deposited into a community repository, provide accession details.* |
| Cell population abundance | *Describe the abundance of the relevant cell populations within post-sort fractions, providing details on the purity of the samples and how it was determined.* |
| Gating strategy | *Describe the gating strategy used for all relevant experiments, specifying the preliminary FSC/SSC gates of the starting cell population, indicating where boundaries between "positive" and "negative" staining cell populations are defined.* |

☐ Tick this box to confirm that a figure exemplifying the gating strategy is provided in the Supplementary Information.

# Magnetic resonance imaging

## Experimental design

| Design type | *Indicate task or resting state; event-related or block design.* |
|---|---|
| Design specifications | *Specify the number of blocks, trials or experimental units per session and/or subject, and specify the length of each trial or block (if trials are blocked) and interval between trials.* |
| Behavioral performance measures | *State number and/or type of variables recorded (e.g. correct button press, response time) and what statistics were used to establish that the subjects were performing the task as expected (e.g. mean, range, and/or standard deviation across subjects).* |

## Acquisition

| Imaging type(s) | *Specify: functional, structural, diffusion, perfusion.* |
|---|---|
| Field strength | *Specify in Tesla* |
| Sequence & imaging parameters | *Specify the pulse sequence type (gradient echo, spin echo, etc.), imaging type (EPI, spiral, etc.), field of view, matrix size, slice thickness, orientation and TE/TR/flip angle.* |
| Area of acquisition | *State whether a whole brain scan was used OR define the area of acquisition, describing how the region was determined.* |

Diffusion MRI    ☐ Used    ☐ Not used

## Preprocessing

| Preprocessing software | *Provide detail on software version and revision number and on specific parameters (model/functions, brain extraction, segmentation, smoothing kernel size, etc.).* |
|---|---|

| Normalization | *If data were normalized/standardized, describe the approach(es): specify linear or non-linear and define image types used for transformation OR indicate that data were not normalized and explain rationale for lack of normalization.* |
|---|---|
| Normalization template | *Describe the template used for normalization/transformation, specifying subject space or group standardized space (e.g. original Talairach, MNI305, ICBM152) OR indicate that the data were not normalized.* |
| Noise and artifact removal | *Describe your procedure(s) for artifact and structured noise removal, specifying motion parameters, tissue signals and physiological signals (heart rate, respiration).* |
| Volume censoring | *Define your software and/or method and criteria for volume censoring, and state the extent of such censoring.* |

## Statistical modeling & inference

| Model type and settings | *Specify type (mass univariate, multivariate, RSA, predictive, etc.) and describe essential details of the model at the first and second levels (e.g. fixed, random or mixed effects; drift or auto-correlation).* |
|---|---|
| Effect(s) tested | *Define precise effect in terms of the task or stimulus conditions instead of psychological concepts and indicate whether ANOVA or factorial designs were used.* |

Specify type of analysis: ☐ Whole brain ☐ ROI-based ☐ Both

| Statistic type for inference<br>(See Eklund et al. 2016) | *Specify voxel-wise or cluster-wise and report all relevant parameters for cluster-wise methods.* |
|---|---|
| Correction | *Describe the type of correction and how it is obtained for multiple comparisons (e.g. FWE, FDR, permutation or Monte Carlo).* |

## Models & analysis

| n/a | Involved in the study |
|---|---|
| ☐ | ☐ Functional and/or effective connectivity |
| ☐ | ☐ Graph analysis |
| ☐ | ☐ Multivariate modeling or predictive analysis |

| Functional and/or effective connectivity | *Report the measures of dependence used and the model details (e.g. Pearson correlation, partial correlation, mutual information).* |
|---|---|
| Graph analysis | *Report the dependent variable and connectivity measure, specifying weighted graph or binarized graph, subject- or group-level, and the global and/or node summaries used (e.g. clustering coefficient, efficiency, etc.).* |
| Multivariate modeling and predictive analysis | *Specify independent variables, features extraction and dimension reduction, model, training and evaluation metrics.* |

