## [Peer Review File · Nature Human Behaviour]

Peer Review Information

Journal: Nature Human Behaviour

Manuscript Title: Testing Theory of Mind in Large Language Models and Humans

Corresponding author name(s): James W. A. Strachan, Cristina Becchio

Reviewer Comments & Decisions:

Decision Letter, initial version:

6th October 2023

Dear Dr Strachan,

Thank you once again for your manuscript, entitled "Testing Theory of Mind in GPT Models and Humans", and for your patience during the peer review process.

Your Article has now been evaluated by 3 referees. You will see from their comments copied below that, although they find your work of potential interest, they have raised quite substantial concerns. In light of these comments, we cannot accept the manuscript for publication, but would be interested in considering a revised version if you are willing and able to fully address reviewer and editorial concerns.

We hope you will find the referees' comments useful as you decide how to proceed. If you wish to submit a substantially revised manuscript, please bear in mind that we will be reluctant to approach the referees again in the absence of major revisions. We are committed to providing a fair and constructive peer-review process. Do not hesitate to contact us if there are specific requests from the reviewers that you believe are technically impossible or unlikely to yield a meaningful outcome.

To guide the scope of the revisions, the editors discuss the referee reports in detail within the team, including with the chief editor, with a view to (1) identifying key priorities that should be addressed in revision and (2) overruling referee requests that are deemed beyond the scope of the current study. We hope that you will find the prioritised set of referee points to be useful when revising your study. Please do not hesitate to get in touch if you would like to discuss these issues further.

1. Reviewer 2 raises important concerns about the newly created tasks. The reviewer asks that you create counterfactual tasks and test the model on it. In addition, they raise concerns about the novel false belief tasks that you created. Specifically, the reviewer is concerned that they do not capture the whole breadth of false belief. We ask that you address these concerns in full.

2. Reviewer 1 has concerns about the replicability of your work. To address this concern, please follow the reviewer's suggestion and replicate your analysis, by incorporating an open source model (for example, a chat version of LLaMA 2).

3. Please address all remaining reviewer concerns in detail. Specifically, we ask that you provide additional methodological details, especially on the variants of the generated tests, and incorporate additional literature on the theory of mind.

If you wish to submit a suitably revised manuscript, we would hope to receive it within 2 months. I would be grateful if you could contact us as soon as possible if you foresee difficulties with meeting this target resubmission date.

- Include a "Response to the editors and reviewers" document detailing, point-by-point, how you addressed each editor and referee comment. If no action was taken to address a point, you must provide a compelling argument. When formatting this document, please respond to each reviewer comment individually, including the full text of the reviewer comment verbatim followed by your response to the individual point. This response will be used by the editors to evaluate your revision and sent back to the reviewers along with the revised manuscript.
- Highlight all changes made to your manuscript or provide us with a version that tracks changes.

[REDACTED]

Thank you for the opportunity to review your work. Please do not hesitate to contact me if you have any questions or would like to discuss the required revisions further.

Sincerely,

[REDACTED]

Reviewer expertise:

Reviewer #1: GPT models

Reviewer #2: theory of mind , intuitive theories

Reviewer #3: GPT models

REVIEWER COMMENTS:

Reviewer #1:

Remarks to the Author:

I read this paper with great interest. The explosion of Large Language Models (LLM) powered conversational AIs has been a momentous event in the development of intelligent machines. These machines have passed many versions of the Turing Test, rendering it almost irrelevant. Yet, many questions remain open. Furthermore, there has been some speculation since Michal Kosinski's pre-print that claimed that 'theory of mind' may have spontaneously emerged in LLMs like ChatGPT. So it is refreshing to see a more systematic attempt at this important question.

I like the paper, and commend the authors on their approach. With that said, I have some suggestions to strengthen the work.

FAUX PAS TEST:

The Faux Pas test plays an important role in the paper, because it is where ChatGPT falls short of human performance. Maybe this is a problem with this reviewer's own theory of mind, but my own intuition was actually in line with ChatGPT: That it was not clear if Uncle Tom knew that the pie was an apple pie. It may simply be that he thought they knew the obvious fact that he didn't like apple pie, but in fact they had forgotten. This exact story happened, in fact, to someone close to me (though with banana cake!). Then, the person who made the cake was still offended, while also recognizing that they made a mistake! So to me, it's highly questionable to use this test as a reliable theory of mind test, penalizing a non-committal answer. Given this, I very much appreciate the approach taken by the authors.

But this indeed highlights the sensitivity of these types of tests. Given this, I would very much appreciate more details about how the authors generated the variants of this (and other) tests. In the methods appendix, the authors stated that they generated additional variants, but do not provide those variants. It would be great if they could provide at least one, accompanying each case.

REPLICABILITY:

A major weakness in the work is about replicability, due to the fact that ChatGPT is both a closed system (without access to model weights), and an evolving system. This means that it would be difficult to replicate the precise analysis, and any further failure to replicate may not be easily attributed. I recognize that there is a trade-off here. On one hand, the authors can use an open source model like LLaMA 2 (more accurately, it should be called 'open weight' since only the weights are open, but not the training algorithm and the entire data used in training). But these models are not considered to be the top performers out there, so it's still valuable to test the most powerful model out

there, namely ChatGPT. I see two possible ways out of this: (1) The authors replicate their analysis, by incorporating an open source model (a chat version of LLaMA 2 for instance). (2) The authors simply describe this as a potential weakness of the results, which may reduce replicability. My preference would be for option (1), assuming the authors have the resources to achieve this. I've seen various online demos of LLaMA-2-Chat (the LLaMA-2 model that's fine tuned for conversation), so this may be rather straightforward. While this does not substantially increase generalizability, it certainly increases replicability.

ORDER EFFECTS:

I appreciate the fact that the authors conducted rigorous analysis of item order effect. But I was wondering why they did not simply just use one chat session per item? Is it in order to mimic the fact that humans, also, do multiple items in a row? This raises a few questions:

- Is there any evidence, from previous work, of order effect in the application of multiple 'theory of mind' tests in humans?
- Why not run single-question-per participant both for humans and for ChatGPT? Wouldn't that provide a cleaner design? And after all, it's particularly cheap to execute with online panels and ChatGPT.

MINOR COMMENTS:

The authors cite recent calls for 'Machine Psychology' that have been put forward by some scholars in experimental psychology. I think it is worthwhile making reference to some slightly earlier work, predating LLMS, on the notion of 'Machine Behavior', in which an interdisciplinary group of scientists (including some psychologists) have called for a broad application of behavioral science methods to AI systems:

Rahwan, I., Cebrian, M., Obradovich, N., Bongard, J., Bonnefon, J. F., Breazeal, C., ... & Wellman, M. (2019). Machine behaviour. *Nature*, 568(7753), 477-486.

On page 2, the authors state that they tested GPT-4 and its predecessor ChatGPT 3.5. This sentence makes it ambiguous whether they used the raw model GPT-4 or the chat-enabled version within ChatGPT 4. I presume it is the latter (i.e. that the authors used the chat version of both models), but it is currently ambiguous.

On the paragraph starting on line 116, the authors describe how they applied the modified version of the Faux Pas and Hinting tests on humans, and noticed that humans performed better on the new test. I think the way this paragraph is written throws off the readers, because I wasn't sure what to make of that statement. Does this mean that the test variant is a bad comparison also for ChatGPT? I later found out (especially through visual inspection of Figure 1A) that the difference in human performance between the two versions of the test is quite small, so this difference does not alter the main qualitative story.

Reviewer #2:

Remarks to the Author:

Summary

This paper presents a comparison between GPT-4, GPT-3.5, and people, on several tasks broadly related to Theory-of-Mind (False belief, hinting, faux pas, irony, strange stories). The authors use well known data-sets and find GPT performs at human-like levels (except for faux pas). The authors then try variations on the well-known data sets and find GPT again performs relatively well (except for faux pas). Some time is spent trying to figure out why the poor performance on faux pas, with a conclusion that prompting for likelihood can reveal a true answer (a lot of the discussion is also spent on the faux pas issue).

There's a lot to like about this paper, and it's certainly useful to have running comparisons of the latest LLM models with various cognitive tasks. At the same time, the level of analysis and comparison seem more suited for a (good) conference than an NHB piece. In particular, there is a lack of stress testing various aspects of the models (already in use in other papers that have come out before this piece), frequent anthropomorphization, and lack of systematic comparison to a wider range of LLMs.

Given the current state of the paper I don't see a way to revise this to bring it in line with NHB. I'm sorry my comments are not more positive, it's not fun to write negative things. I do hope the authors use the comments below to make the paper stronger elsewhere, as I do think it should be out there for discussion in community.

Comments

1) This is more minor, but I honestly don't find it terribly useful to show that GPT-4 can pass widely available tasks that have been in the literature for decades. The authors themselves accept this when they raise the possibility that GPT is only passing the tasks because it already trained on them, and so the need for 'new' tasks. But this is presented as though the main thing is that they pass the old ones, and we're just doing a kind of sanity check by trying new ones as well, which it also passes, so all good. It should more be 'it's not really that surprising that it can pass tasks in the literature, that's more of a sanity check, the main thing is new tasks'. This is more an issue of framing but it becomes important in what comes next.

2) Following on the previous, given the importance people currently associate with the whole question of whether LLMs are 'simply' finding statistical patterns in the massive amounts of data they've seen or learning more human-like models, it seems like the right thing to do is to generate what some are calling 'counterfactual' tasks that keep the spirit of the original but vary it in meaningful ways that cannot be trivially passed by pattern matching.

There is a difficulty in the current literature in saying how exactly to define such tasks, it's currently more a 'know it when you see it'. Certainly we can agree that taking the Sally-Ann task and changing 'Sally' to 'Wally' and 'Ann' to 'Jann' (but keeping everything else the same) does not count. The "novel" false belief stories are more complicated than that, but they still seem to hew pretty closely to the basic formulation of

'there are two characters, X and Y. X put A in location 1 and left. Y took A from location 1 and put it in location 2. X came back. Where will X look for A?'

Theory of mind in general, and false belief in particular, is broader than this formulation, and yet all

the 'novel' false belief tasks used by the authors seem to follow this pattern. This is the importance stressed by different researchers (including some cited by the current authors), who have come up with different formulations and variations which do seem to throw off the current models (see comments on some missing citations as well below).

To take one example, consider the following different take on one of the prompts:

"Anna and Beth are going to a concert together. Anna has the tickets, and she puts them in the side pocket of her purse and then leaves her purse with Beth while she goes to the bathroom.

While Anna is gone, Beth thinks about getting out the tickets to check their seats. Anna comes back from the bathroom. Beth takes the tickets out of the side pocket and puts the tickets back in the front pocket of Anna's purse.

When they go to the venue, where will Anna check for the tickets?"

GPT-4: Anna will check the side pocket of her purse for the tickets because that's where she originally put them.

Note that Beth did not actually take the tickets out when Anna was away, she just thought of it. Side note, I'm guessing this formulation is likely to trip up developmentalists who have seen a million Salley-Ann tasks and are skimming by the text. Similar variations like a see-through purse also trip up GPT-4.

One can perhaps complain that the authors used many runs of GPT-3.5 and GPT-4 and many tasks and I'm simply using one change, but this change is based on existing papers that have pointed this stuff out. It's fine for the authors to disagree with the previous lit on it but they mainly just don't seem to be engaging with it. I'm thinking here also of "Clever Hans or Neural Theory of Mind? Stress Testing Social Reasoning in Large Language Models" and "How Well Do Large Language Models Perform on Faux Pas Tests?".

More generally this is a lack of clarity in the main paper for how the 'novel' items were generated, by what criteria, etc.

3) Beyond the false-belief tasks which don't really engage with the stress testing emphasized by prev lit, I found the faux pas tasks somewhat confusing. It seems like the right answer is always 'yes'? I understand this may be an issue in the human-literature as well but it seems like a better test might be to have prompts where the correct answer is sometimes yes and sometimes no, otherwise a model biased to answer yes will simply pass things?

Consider that in trying the following (no context beyond the prompt below) with GPT-4 15 times:

"in the story, did someone say something that they should not have?"

you can only answer 'yes' or 'no', restrict your answers to either one of those words"

The GPT-4 answer is pretty much always "yes"

In a follow up to the above, if asked again without context:

"is it more likely that Joe knew that Mike was in the cubicles?"

you can only answer 'yes' or 'no', restrict your answers to either one of those words"

GPT-4 pretty much always answered 'yes'.

Given this initial yes-bias it is difficult to conclude from getting GPT-4 to say 'yes' on tasks where the right answer is always 'yes' that it deeply understands faux pas.

4) While I understand the focus on GPT-3.5 and GPT-4 these are closed off models, there are many currently available LLMs and it seems like a comparison for what they can and can't do in terms of training, vs. size, vs. RLHF yes-or-no, vs. other model parameters could really be better served by a more broad comparison. Again, this is fine for a conference submission, but seems less right for NHB.

5) minor, but it feels kind of wrong to cite Sap et al. (line 60) as though it backs up the idea that current LLMs are good at ToM when they say "Our results show that models struggle substantially at these Theory of Mind tasks"

6) The opening seems relatively reference-free and does not engage in a bunch of the previous literature on what theory-of-mind is or isn't. I'm not expecting a review piece but it again seems to fall short of the standards of NHB.

7) Very minor: "It is possible that such processes could lead to an overly conservative approach to drawing conclusions from incomplete information where GPT models do not commit to the likeliest explanation despite being able to recognise it." -- if this were true then GPT-4 would not answer *any* pragmatics tasks (including the hinting ones) as they require a reading from incomplete information. Even in the opening examples of 'it sure is hot here!' implying 'I am politely asking that you open the window' require a reading in of additional information.

Reviewer #3:

Remarks to the Author:

This is an interesting paper, I just have two comments:

1) I disagree with the last sentence of the abstract. More importantly I don't see how your findings support this. It seems more like a belief.

2) The tone of the abstract overall is surprisingly lukewarm. From the experiment it seems that GPT-4 is absolutely at human level. The only issue with faux pas is the safety fine-tuning which makes it extra cautious, but this is not "truly" an issue with GPT-4, more with the ChatGPT interface. I think the

abstract should represent faithfully that your experiments conclusively demonstrate theory of mind capabilities at human level for GPT-4 (which is also my own conclusion after a lot of testing on GPT-4).

Author Rebuttal to Initial comments

REVIEWER COMMENTS

Reviewer #1

I read this paper with great interest. The explosion of Large Language Models (LLM) powered conversational AIs has been a momentous event in the development of intelligent machines. These machines have passed many versions of the Turing Test, rendering it almost irrelevant. Yet, many questions remain open. Furthermore, there has been some speculation since Michal Kosinski's pre-print that claimed that 'theory of mind' may have spontaneously emerged in LLMs like ChatGPT. So it is refreshing to see a more systematic attempt at this important question.

I like the paper, and commend the authors on their approach. With that said, I have some suggestions to strengthen the work.

We thank the Reviewer for their insights, which helped to improve the paper substantially.

1. The Faux Pas test plays an important role in the paper, because it is where ChatGPT falls short of human performance. Maybe this is a problem with this reviewer's own theory of mind, but my own intuition was actually in line with ChatGPT: That it was not clear if Uncle Tom knew that the pie was an apple pie. It may simply be that he thought they knew the obvious fact that he didn't like apple pie, but in fact they had forgotten. This exact story happened, in fact, to someone close to me (though with banana cake!). Then, the person who made the cake was still offended, while also recognizing that they made a mistake! So to me, it's highly questionable to use this test as a reliable theory of mind test, penalizing a non-committal answer. Given this, I very much appreciate the approach taken by the authors.

***RL1.** Individual intuitions can show a great detail of variability across human samples — particularly for scenarios that may have parallels to some participants' lived experiences. When writing the original manuscript, we selected an example faux pas item for illustration that, upon further inspection, turned out to be highly variable in human intuitions (average human performance was at 78.40%, the second lowest score on any individual item). To avoid confounding intuitions we have changed the example in the text to use one of the items that*

showed a high human consensus (92.16%), which we include below for the reviewer's convenience:

Jill had just moved into a new house. She went shopping with her Mum and bought some new curtains. When Jill had just put them up, her best friend Lisa came round and said, "Oh, those curtains are horrible, I hope you're going to get some new ones." Jill asked, "Do you like the rest of my bedroom?"

The variability in participants' intuitions to these scenarios makes it all the more important to collect data from large human samples to be compared with the responses of LLMs, and to use the same testing protocols across human and non-human responders. We now present a full item-level breakdown of responses on each test in Supplementary Figure S2, detailing the average performance and relative variability (coefficient of variation) in item-wise performance.

2. But this indeed highlights the sensitivity of these types of tests. Given this, I would very much appreciate more details about how the authors generated the variants of this (and other) tests. In the methods appendix, the authors stated that they generated additional variants, but do not provide those variants. It would be great if they could provide at least one, accompanying each case.

RI.2. *In the original version of the manuscript, we included novel items that matched the logic and structure of the original items but differed in terms of detail and semantic content. This was to ensure that models did not simply replicate training set data. We have clarified this on p.3 of the revised text. Following the feedback of the reviewers, we now distinguish these novel items from new variants of the False Belief (False Belief Perturbations, see Supplementary Section 4) and Faux Pas tests (Faux Pas Likelihood Test, p.6-8, and Belief Likelihood Test, p.8-10). These new variants manipulate the original logic and structure of the tests in order to serve as controls or test alternative explanations. In the interest of clarity, we now refer to new items that follow the same structure and logic as the original test as "novel items" and new variants of the tests that modify the structure and logic as perturbations (False Belief) and variants (Faux Pas) throughout the manuscript. The text of original and novel items and the coded responses for tests in the Theory of Mind Battery and novel variants are available on the OSF (see Methods, Resource Availability)*

3. A major weakness in the work is about replicability, due to the fact that ChatGPT is both a closed system (without access to model weights), and an evolving system. This means that it would be difficult to replicate the precise analysis, and any further failure to replicate may not be easily attributed. I recognize that there is a trade-off here. On one hand, the authors can use an open source model like LLaMA 2 (more accurately, it should be called 'open weight' since only the

weights are open, but not the training algorithm and the entire data used in training). But these models are not considered to be the top performers out there, so it's still valuable to test the most powerful model out there, namely ChatGPT. I see two possible ways out of this: (1) The authors replicate their analysis, by incorporating an open source model (a chat version of LLaMA 2 for instance). (2) The authors simply describe this as a potential weakness of the results, which may reduce replicability. My preference would be for option (1), assuming the authors have the resources to achieve this. I've seen various online demos of LLaMA-2-Chat (the LLaMA-2 model that's fine tuned for conversation), so this may be rather straightforward. While this does not substantially increase generalizability, it certainly increases replicability.

RL.3. *Our decision to focus on ChatGPT was driven by the prominence of GPT-4 as the state-of-the-art in language models. However, we fully agree that the closed and evolving nature of ChatGPT is a valid concern and we have taken efforts to address this in our revised manuscript. Following the reviewer's suggestion, we have substantially extended our dataset to include responses from three LLaMA-2-Chat models (70B, 13B, and 7B). We report the results of the 70B model in the main manuscript, and the results of the other two models are included in the Supplementary Section 1.*

As shown in the revised Figure 1 (included below for the reviewer's convenience), the LLaMA2-70B model showed a different pattern of results from the two GPT models: while LLaMA2-70B performed significantly worse than human levels on the Irony, the Hinting, and the Strange Stories tests, it performed significantly better than humans on the Faux Pas.

Figure 1. Performance of human (purple), GPT-4 (dark blue), GPT-3.5 (light blue), and LLaMA2-70B (green) on the battery of Theory of Mind tests. **A.** Violin plot on original test items for each test showing the distribution of test scores for individual sessions and participants. Coloured dots show the average of the response score across all test items for each individual test session (LLMs) or participant (humans). Black dots indicate the median for each condition. Significance markers show the results of Holm-corrected Wilcoxon two-way tests comparing LLM scores against human scores. Tests are ordered in descending order of human performance. **B.** Barbell plot showing interquartile ranges of the average scores on the original published items (dark colours) and novel items (pale colours) across each test. Diamonds indicate the median scores. Significance markers show the results of Holm-corrected Wilcoxon two-way tests

comparing performance on original items against the novel items generated as controls for this study (* $p < .05$; ** $p < .01$; *** $p < .001$).

This contrast of LLaMA2-70B with GPT-4 motivated us to develop new Faux Pas test variants (Belief Likelihood Test, section ‘Testing information integration’ p.9) that manipulated the inference of specific mental state content by changing the key utterance (which in the original task implied the speaker did not know some key context) to either imply the speaker did know the context or to a neutral statement that did not imply one way or another. Our findings indicate that, while GPT models and humans were sensitive to the implied mental content of all three variants, LLaMA2-70B’s responses were not, indicating a possible response bias or ‘Clever Hans’ type explanation.

Testing open-weight models allows for better reproducibility of our results, but it does not entirely overcome the limitations of testing closed, evolving models such as ChatGPT which show specific patterns of responses that other LLMs do not emulate. We acknowledge this as an important consideration for future work in the new final paragraph in the discussion (p.12).

4. I appreciate the fact that the authors conducted rigorous analysis of item order effect. But I was wondering why they did not simply just use one chat session per item? Is it in order to mimic the fact that humans, also, do multiple items in a row? This raises a few questions:
 - Is there any evidence, from previous work, of order effect in the application of multiple ‘theory of mind’ tests in humans?

RL.4a. *To the best of our knowledge, such item-level order effects have not been studied in previous work with humans primarily because it is unlikely that humans would learn how to solve these tasks over the course of a single administration. Humans, particularly neurotypical adults, are expected to solve these tasks using higher-order social cognitive abilities rather than picking up on statistical regularities within the text (such as naming the first location mentioned in the False Belief task). Given the probabilistic nature of LLMs, however, the possibility that apparent social reasoning could be explained by lower level processes is a more serious concern, which was what motivated our analysis of item order effects. We have streamlined the presentation of these analyses, which in the interest of space are now reported in Supplementary Section 3.*

- Why not run single-question-per participant both for humans and for ChatGPT? Wouldn’t that provide a cleaner design? And after all, it’s particularly cheap to execute with online panels and ChatGPT.

RI.4b. *The Theory of Mind tests presented in the battery, with the exception of the Irony comprehension test, are validated social psychological tests in humans. The choice of providing all items of a test to each participant/session was intended to preserve content validity (i.e. how well a test covers all relevant parts of the construct it aims to measure).*

Our revised manuscript includes the collection of new data both with LLMs and human participants on newly designed control variants of the Faux Pas tests (main text) and False Belief perturbations (Supplementary Section 4). Given that concerns of preserving construct validity do not apply to these new tests, we have incorporated the reviewer’s suggestion and administered a single-question-per-run approach for both human and LLM data. We report this information, together with a description of these control variants, on p.5 of the main text and in Supplementary Section 4 (False Belief perturbations), p.9-10 of the main text (Faux Pas variants, Belief Likelihood Test), and p.20 of the main text (Methods, Belief Likelihood Test).

5. The authors cite recent calls for ‘Machine Psychology’ that have been put forward by some scholars in experimental psychology. I think it is worthwhile making reference to some slightly earlier work, predating LLMS, on the notion of ‘Machine Behavior’, in which an interdisciplinary group of scientists (including some psychologists) have called for a broad application of behavioral science methods to AI systems: Rahwan, I., Cebrian, M., Obradovich, N., Bongard, J., Bonnefon, J. F., Breazeal, C., ... & Wellman, M. (2019). Machine behaviour. *Nature*, 568(7753), 477-486.

RI.5. *We thank the reviewer for bringing this reference to our attention. This reference is indeed highly relevant to our approach. We now cite it (Ref [28]) in the Introduction on p.2.*

6. On page 2, the authors state that they tested GPT-4 and its predecessor ChatGPT 3.5. This sentence makes it ambiguous whether they used the raw model GPT-4 or the chat-enabled version within ChatGPT 4. I presume it is the latter (i.e. that the authors used the chat version of both models), but it is currently ambiguous.

RI.6. *We apologise for the confusion and thank the reviewer for highlighting this. We did indeed use the chat-enabled version of both models. We have clarified this in the text on p.2.*

7. On the paragraph starting on line 116, the authors describe how they applied the modified version of the Faux Pas and Hinting tests on humans, and noticed that humans performed better on the new test. I think the way this paragraph is written throws off the readers, because I wasn’t sure what to make of that statement. Does this mean that the test variant is a bad comparison also for ChatGPT? I later found out (especially through visual inspection of Figure 1A) that the difference

in human performance between the two versions of the test is quite small, so this difference does not alter the main qualitative story.

R1.7. *The reported differences in human performance are indeed small and do not alter the conclusions. We tested the difference on human performance for original vs. novel items in order to be sure that if (e.g.) GPT did perform significantly worse on the novel items that we generated that this would not be due to those vignettes just being much more difficult to pass. Small differences in the average performance on original and novel vignettes are to be expected given the variability on most tests and differences in the number of original and novel vignettes. We have restructured the presentation of our results to better contextualise the analysis of original vs. novel items (p.4-6)*

Reviewer #2

1. This paper presents a comparison between GPT-4, GPT-3.5, and people, on several tasks broadly related to Theory-of-Mind (False belief, hinting, faux pas, irony, strange stories). The authors use well known data-sets and find GPT performs at human-like levels (except for faux pas). The authors then try variations on the well-known data sets and find GPT again performs relatively well (except for faux pas). Some time is spent trying to figure out why the poor performance on faux pas, with a conclusion that prompting for likelihood can reveal a true answer (a lot of the discussion is also spent on the faux pas issue).

There's a lot to like about this paper, and it's certainly useful to have running comparisons of the latest LLM models with various cognitive tasks. At the same time, the level of analysis and comparison seem more suited for a (good) conference than an NHB piece. In particular, there is a lack of stress testing various aspects of the models (already in use in other papers that have come out before this piece), frequent anthropomorphization, and lack of systematic comparison to a wider range of LLMs.

Given the current state of the paper I don't see a way to revise this to bring it in line with NHB. I'm sorry my comments are not more positive, it's not fun to write negative things. I do hope the authors use the comments below to make the paper stronger elsewhere, as I do think it should be out there for discussion in community

R2.1. *We are grateful to the Reviewer for bringing up important points and for providing useful suggestions, which helped us improving the paper.*

*Stress testing: We acknowledge the importance of stress testing in evaluating model robustness and realise that this aspect was not sufficiently addressed in our initial submission. As detailed in ***R2.3.***, we have now incorporated new control variants that evaluate LLMs' performance under*

various challenging scenarios and validate these variants with human participants. These variations include some of those used in recent publications that the reviewer mentions.

Anthropomorphisation: This was intended to simplify the description of the results. However, we agree that such language can potentially lead to misunderstandings about the capabilities and nature of these language models. We have revised the text of the manuscript to eliminate anthropomorphic terms. In addition, we now specify that we refer to inference here not in the sense of the processes by which biological organisms infer hidden states from their environment, but rather as any process of reasoning whereby conclusions are derived from a set of propositional premises.

Range of LLMs: Our primary aim was to compare GPT model performance with that of humans, rather than conducting an extensive comparison with a wider range of LLMs. This is because although several previous studies compare a broader selection of LLMs, few have incorporated human data collected under comparable conditions. Nevertheless, we acknowledge the significance of comparing with open-weight models to enhance reproducibility and have now incorporated data from three additional models, specifically the 70B, 13B, and 7B versions of LLaMA2-Chat.

2. This is more minor, but I honestly don't find it terribly useful to show that GPT-4 can pass widely available tasks that have been in the literature for decades. The authors themselves accept this when they raise the possibility that GPT is only passing the tasks because it already trained on them, and so the need for 'new' tasks. But this is presented as though the main thing is that they pass the old ones, and we're just doing a kind of sanity check by trying new ones as well, which it also passes, so all good. It should more be 'it's not really that surprising that it can pass tasks in the literature, that's more of a sanity check, the main thing is new tasks'. This is more an issue of framing but it becomes important in what comes next.

R2.2. *The Theory of Mind tests presented in the battery, with the exception of the Irony comprehension test, are validated social psychological tests in humans. These have been shown to have high internal validity and high reliability for measuring human social cognitive skills. These tests therefore provide a reliable benchmark for measuring LLMs against human performance, and using these validated tests is all the more important given recent claims that LLMs demonstrate a human-like capacity for Theory of Mind that rely on the use of single tests or even single items.*

The reviewer's comment that it is not terribly useful to show that GPT-4 can pass these tasks implies that GPT-4 does, in fact, pass and this is nothing more than a sanity check, but this is clearly not the case. GPT-4 does not pass all tests: despite high performance on False Belief,

Hinting, Strange Stories and Irony (although this is not a published test), GPT-4 fails on Faux Pas. This is indeed surprising.

That being said, we agree that the use of validated tests ought to be complemented by the development of new items and variants. In response to the feedback of the reviewer, we have collected additional data from all LLMs and a new sample of human participants addressing perturbations of the False Belief (see R2.3.). Additionally, we have developed and tested a new set of variants of the Faux Pas (the Belief Likelihood Test; see R2.7b.). These new studies are presented on p.8-10 of the revised manuscript (Belief Likelihood Test in section 'Testing information integration') and p.5 and Supplementary Section 4 (False Belief Perturbations).

- Following on the previous, given the importance people currently associate with the whole question of whether LLMs are 'simply' finding statistical patterns in the massive amounts of data they've seen or learning more human-like models, it seems like the right thing to do is to generate what some are calling 'counterfactual' tasks that keep the spirit of the original but vary it in meaningful ways that cannot be trivially passed by pattern matching.

There is a difficulty in the current literature in saying how exactly to define such tasks, it's currently more a 'know it when you see it'. Certainly we can agree that taking the Sally-Ann task and changing 'Sally' to 'Wally' and 'Ann' to 'Jann' (but keeping everything else the same) does not count. The "novel" false belief stories are more complicated than that, but they still seem to hew pretty closely to the basic formulation of 'there are two characters, X and Y. X put A in location 1 and left. Y took A from location 1 and put it in location 2. X came back. Where will X look for A?'

Theory of mind in general, and false belief in particular, is broader than this formulation, and yet all the 'novel' false belief tasks used by the authors seem to follow this pattern. This is the importance stressed by different researchers (including some cited by the current authors), who have come up with different formulations and variations which do seem to throw off the current models (see comments on some missing citations as well below).

To take one example, consider the following different take on one of the prompts:

"Anna and Beth are going to a concert together. Anna has the tickets, and she puts them in the side pocket of her purse and then leaves her purse with Beth while she goes to the bathroom.

While Anna is gone, Beth thinks about getting out the tickets to check their seats. Anna comes back from the bathroom. Beth takes the tickets out of the side pocket and puts the tickets back in the front pocket of Anna's purse.

When they go to the venue, where will Anna check for the tickets?"

GPT-4: Anna will check the side pocket of her purse for the tickets because that's where she originally put them.

Note that Beth did not actually take the tickets out when Anna was away, she just thought of it. Side note, I'm guessing this formulation is likely to trip up developmentalists who have seen a million Salley-Ann tasks and are skimming by the text. Similar variations like a see-through purse also trip up GPT-4.

R2.3. *We appreciate the suggestion to test LLMs on control variants of the False Belief task (we reserve the term 'counterfactual' for referring to the form of hypothetical reasoning used to solve faux pas, e.g., "Had they known, they would not have said X"), and have incorporated this by collecting a new dataset where we tested LLMs (GPT-4, GPT-3.5, and LLaMA2-70B) on the four unexpected-transfer control variants put forward by Ullman (2023). We administered perturbations of three different stories, in order to mitigate the limitations of a one-shot investigation, generating 15 new items in total.*

Importantly, we also administered these control variants to human subjects (N=50 for each item; total N=750). An untested assumption is that the proposed variants implement "small perturbations that shouldn't matter to an entity that has ToM" (quoting Ullman, 2023). However, as the reviewer points out, these formulations could also trip up humans in ways that have nothing to do with Theory of Mind but instead lead to misreading or misinterpretation. Indeed, while we replicated the finding that GPT-4 and other LLMs performed poorly on these control variants, humans also failed on half of these items. We described these results in Supplementary Section 4 and refer to them on p.5 of the revised manuscript. We include Supplementary Figure S4, which summarises these results, below for the reviewer's convenience.

Figure S4. Performance of LLMs and humans across perturbations of the False Belief task.

As discussed in Supplementary Section 4, it is worth noting that these control variants present diverse challenges for LLMs that go beyond tracking mental states, and involve understanding physical properties, relationships between objects, and spatial reasoning capabilities that have nothing to do with Theory of Mind. They also differ in terms of the type of belief updating: the variants where humans performed ‘poorly’ according to the intuitions proposed by Ullman are those where the character’s belief can only be updated after they return to the room, while other variants where performance is more successful involve manipulations of belief states that exist prior to returning. Methodologically, these results highlight the importance of validating test variants with human participants and conducting systematic investigations that do not conflate what humans ‘ought to’ find trivial with what is actually trivial.

- One can perhaps complain that the authors used many runs of GPT-3.5 and GPT-4 and many tasks and I’m simply using one change, but this change is based on existing papers that have pointed this stuff out. It’s fine for the authors to disagree with the previous lit on it but they mainly just don’t seem to be engaging with it. I’m thinking here also of “Clever Hans or Neural Theory of Mind? Stress Testing Social Reasoning in Large Language Models” and “How Well Do Large Language Models Perform on Faux Pas Tests?”.

R2.4. *We are grateful for these references. We have included them when introducing the control variants for the False Belief and Faux Pas tests. These references are numbers [27] and [36] respectively, and are cited for the first time on p.2 and p.6 of the revised manuscript.*

5. More generally this is a lack of clarity in the main paper for how the ‘novel’ items were generated, by what criteria, etc.

R2.5. *In the original version of the manuscript, we included novel vignettes that matched the logic and structure of the original items but differed in terms of detail and semantic content. This was to control for potential familiarity with the published text of the questions. We have clarified this on p.3 of the revised text. Following the feedback of the reviewers, we now distinguish these novel items from new variants of the False Belief (False Belief Perturbations, see Supplementary Section 4) and Faux Pas tests (Faux Pas Likelihood Test, p.6-8, and Belief Likelihood Test, p.8-10). These new variants manipulate the original logic and structure of the tests in order to serve as controls or test alternative explanations. In the interest of clarity, we now refer to new items that follow the same structure and logic as the original test as “novel items” and new variants of the tests that modify the structure and logic as perturbations (False Belief) and variants (Faux Pas) throughout the manuscript.*

6. Beyond the false-belief tasks which don’t really engage with the stress testing emphasized by prev lit, I found the faux pas tasks somewhat confusing. It seems like the right answer is always ‘yes’?

R2.6. *In our original manuscript, all Faux Pas stories were delivered with four questions. The first question was always, “Did somebody say something they should not have said?”, and the correct answer to this question is always yes. The second question asked the respondent to report what the person said that they should not have said, and the third question was a comprehension question specific to the content of the story. The fourth and key question, which was our focus for coding, asked specifically about the character’s knowledge or belief state when they made the utterance; “Did [Lisa] know [the curtains were new]?” The correct answer to this final question was always no.*

We have taken effort to clarify this in our description of the Faux Pas test, both when describing the results (p.6) and the methods (p.16-17).

7. I understand this may be an issue in the human-literature as well but it seems like a better test might be to have prompts where the correct answer is sometimes yes and sometimes no, otherwise a model biased to answer yes will simply pass things?
Consider that in trying the following (no context beyond the prompt below) with GPT-4 15 times:

“in the story, did someone say something that they should not have?
you can only answer ‘yes’ or ‘no’, restrict your answers to either one of those words”
The GPT-4 answer is pretty much always “yes”

R2.7a. *Since our coding was based on the answers to the final (belief) question, any bias would have to affect the responses to this question only (although see Supplementary Section 5 for a recoding that considers responses to other questions). A potential yes-bias is unlikely to explain the pattern of results that we observe. If the poor performance of GPT models on the original “Did the person know...” were affected by a yes-bias, then we would expect the majority of errors to be due to models’ responding that the person did know. However, only 2 responses that were coded as errors (out of a total 349 on the whole task) were affirmative. The rest of the responses were noncommittal answers (e.g. “There is not enough information to tell”). Importantly, such noncommittal responses were virtually non-existent in human responses. We now report this information on p.6-7 of the revised manuscript.*

In a follow up to the above, if asked again without context:

“is it more likely that Joe knew that Mike was in the cubicles?
you can only answer ‘yes’ or ‘no’, restrict your answers to either one of those words”
GPT-4 pretty much always answered ‘yes’.

Given this initial yes-bias it is difficult to conclude from getting GPT-4 to say ‘yes’ on tasks where the right answer is always 'yes' that it deeply understands faux pas.

R2.7b. *The original manuscript contained a misprint in Figure 1C (Figure 2A of the revised manuscript) where the likelihood question was inadvertently truncated. The correct question should have been, “Is it more likely that the person knew or did not know...?” We apologise for any misunderstanding this has caused. This is crucial because a simple “yes” response does not fit the question here, thus ruling out a yes-bias explanation.*

However, the reviewer’s point about a potential bias influencing model performance on this type of question is valid. Specifically, there is a deeper concern about a possible bias explaining the better performance of the GPT models on the likelihood framing of the faux pas question, and the high performance of LLaMA2-70B on both versions: if models are generally biased towards attributing a lack of knowledge to characters (a did-not-know bias), this could create an illusion of understanding faux pas.

To control for this Clever Hans-type effect, we ran a follow-up study manipulating the information encoded in the speaker’s utterance to test if participants (LLM and human) were truly sensitive to the belief information encoded within the vignette. For each story, we created three variants: a Faux Pas variant, a Neutral variant, and a Knowledge-Implied variant. In the Faux

Pas, the utterance implied that the speaker did not know the context, in the Neutral variant, the utterance suggested neither that they knew nor did not know, and in the Knowledge-Implied variant the utterance implied that they did know. If the models' responses reflect a true differentiation of the likelihood that the speaker knew or did not know on the basis of information encoded within the story, then we should see a bilateral shift in their responses between variants. That is, relative to the Neutral condition in which models should be more or less unbiased towards saying the person knew or did not know, in the Knowledge Implied variant they should be more likely to report that the speaker knew, and in the Faux Pas variant they should be more likely to report that the speaker did not know. In contrast, if the models' responses were driven purely by a response bias, their responses should not differentiate between these three variants.

Our findings with this follow-up study (included below for the reviewer's convenience in panel B) indicate that GPT models' responses are able to differentiate the information encoded in the story and they show human-like shifts towards attributing knowledge or ignorance depending on what the story implies. LLaMA2-70B, on the other hand, is more likely to attribute ignorance to a Faux Pas variant than a Neutral variant but does not differentiate between Neutral and Knowledge-Implied, suggesting that its responses do not reflect a true sensitivity to the likelihoods of alternative explanations and the information encoded within the stories. We describe this follow-up and the results starting on p.8.

Figure 2. Results of the variants of the Faux Pas test. **A.** Repeated-measures raincloud plot showing the performance of the two GPT models on the original framing of the faux pas question (“Did they know...?”) and the likelihood framing (“Is it more likely that they knew or didn’t know...?”). Points show average score across trials on particular vignette items to allow comparison between the original Faux Pas test and the new Faux Pas Likelihood test. **B.** Bar plot showing the averaged response codes to the likelihood question across the Faux Pas (pink), Neutral (grey) and Knowledge-Implied variants (teal). “Didn’t know” responses are assigned -1, “Knew” responses are assigned +1, and equivocating or unsure responses are

assigned 0. Negative (left) bars indicate a bias towards saying the person did not know the context, and positive (right) bars indicating a bias towards saying they knew. Error bars show the 95% binomial confidence intervals. Significance markers show the results of Holm-corrected Wilcoxon two-way tests comparing the Faux Pas and Knowledge-Implied conditions against Neutral. (** $p < .01$; *** $p < .001$)

8. While I understand the focus on GPT-3.5 and GPT-4 these are closed off models, there are many currently available LLMs and it seems like a comparison for what they can and can't do in terms of training, vs. size, vs. RLHF yes-or-no, vs. other model parameters could really be better served by a more broad comparison. Again, this is fine for a conference submission, but seems less right for NHB.

R2.8. We agree with the reviewer and have extended our dataset to include responses from three LLaMA2-Chat models (70B, 13B, and 7B). We report the results of the 70B model in the main manuscript, and the results of the 13B and 7B models in the Supplementary Section 1.

9. minor, but it feels kind of wrong to cite Sap et al. (line 60) as though it backs up the idea that current LLMs are good at ToM when they say "Our results show that models struggle substantially at these Theory of Mind tasks"

R2.9. We recognise that the way this reference was previously cited was prone to misinterpretation and have endeavoured to clarify our wording. Given our revisions of the introduction, this paragraph has been revised more extensively on p.2. Under the updated reference numbering, Sap et al. is reference [23].

10. The opening seems relatively reference-free and does not engage in a bunch of the previous literature on what theory-of-mind is or isn't. I'm not expecting a review piece but it again seems to fall short of the standards of NHB.

R2.10. As suggested, we have expanded the introduction to include a brief overview of the literature on Theory of Mind with a particular focus on key reviews, theoretical stances, and empirical papers that directly inform the current study. This revised text begins on p.2 of the revised manuscript.

11. Very minor: "It is possible that such processes could lead to an overly conservative approach to drawing conclusions from incomplete information where GPT models do not commit to the likeliest explanation despite being able to recognise it." -- if this were true then GPT-4 would not answer **any** pragmatics tasks (including the hinting ones) as they require a reading from incomplete information. Even in the opening examples of 'it sure is hot here!' implying 'I am politely asking that you open the window' require a reading in of additional information.

R2.11. *We agree with the reviewer that both Faux Pas and Hinting require building on incomplete information. One relevant difference is that for the Hinting task, responders are prompted to speculate, which allows for an open-ended generation of text that LLMs are well suited to. Faux Pas, on the other hand, requires not only speculation (generating information beyond that included in the text) but also using that speculation to commit to a conclusion. Committing is a multi-stage process that is a requisite for using Theory of Mind as a way to predict the actions of others, and this dissociation mirrors some recent work that indicates that the default GPT-4 model particularly struggles at relating speculation about characters' mental states into strategies for action (Zhou et al., <https://arxiv.org/abs/2310.03051>). We now clarify this distinction in the text of the Discussion on p.11*

Reviewer #3

This is an interesting paper, I just have two comments:

We thank the reviewer for these kind words and for the suggestions for improving the Abstract.

1. I disagree with the last sentence of the abstract. More importantly I don't see how your findings support this. It seems more like a belief.

R3.1. *We agree and have revised the sentence to reflect the results of our study.*

2. The tone of the abstract overall is surprisingly lukewarm. From the experiment it seems that GPT-4 is absolutely at human level. The only issue with faux pas is the safety fine-tuning which makes it extra cautious, but this is not "truly" an issue with GPT-4, more with the ChatGPT interface. I think the abstract should represent faithfully that your experiments conclusively demonstrate theory of mind capabilities at human level for GPT-4 (which is also my own conclusion after a lot of testing on GPT-4).

R3.2. *The reviewer is correct in pointing out that the experiments included in the original submission demonstrate that (i) GPT-4 passes validated Theory of Mind tests used to assess social cognitive capacities in humans; (ii) performance on the Faux Pas reflects an excessively conservative stance rather than a limitation of mental reasoning abilities. In response to feedback from the reviewers and editors, we have now also included variations of the Faux Pas that rule out a Clever Hans-type effect in GPT-4. We have thoroughly revised the abstract to emphasise these points.*

Decision Letter, first revision:

16th February 2024

Dear Dr. Strachan,

Thank you for your patience as we've prepared the guidelines for final submission of your Nature Human Behaviour manuscript, "Testing Theory of Mind in Large Language Models and Humans" (NATHUMBEHAV-23082621A). Please carefully follow the step-by-step instructions provided in the attached file, and add a response in each row of the table to indicate the changes that you have made. Please also address the additional marked-up edits we have proposed within the reporting summary. Ensuring that each point is addressed will help to ensure that your revised manuscript can be swiftly handed over to our production team.

We would hope to receive your revised paper, with all of the requested files and forms within two-three weeks. Please get in contact with us if you anticipate delays.

Nature Human Behaviour offers a Transparent Peer Review option for new original research manuscripts submitted after December 1st, 2019. As part of this initiative, we encourage our authors to support increased transparency into the peer review process by agreeing to have the reviewer comments, author rebuttal letters, and editorial decision letters published as a Supplementary item. When you submit your final files please clearly state in your cover letter whether or not you would like to participate in this initiative. Please note that failure to state your preference will result in delays in accepting your manuscript for publication.

In recognition of the time and expertise our reviewers provide to Nature Human Behaviour's editorial process, we would like to formally acknowledge their contribution to the external peer review of your manuscript entitled "Testing Theory of Mind in Large Language Models and Humans". For those reviewers who give their assent, we will be publishing their names alongside the published article.

Cover suggestions

We welcome submissions of artwork for consideration for our cover. For more information, please see our guide for cover artwork.

ORCID

Non-corresponding authors do not have to link their ORCID but are encouraged to do so. Please note that it will not be possible to add/modify ORCIDs at proof. Thus, please let your co-authors know that if they wish to have their ORCID added to the paper they must follow the procedure described in the following link prior to acceptance:

Nature Human Behaviour has now transitioned to a unified Rights Collection system which will allow our Author Services team to quickly and easily collect the rights and permissions required to publish your work. Approximately 10 days after your paper is formally accepted, you will receive an email in providing you with a link to complete the grant of rights. If your paper is eligible for Open Access, our Author Services team will also be in touch regarding any additional information that may be required to arrange payment for your article.

Please note that *Nature Human Behaviour* is a Transformative Journal (TJ). Authors may publish their research with us through the traditional subscription access route or make their paper immediately open access through payment of an article-processing charge (APC). Authors will not be required to make a final decision about access to their article until it has been accepted. Find out more about Transformative Journals

[REDACTED]

Best regards,

[REDACTED]

On behalf of

[REDACTED]

Reviewer #1:

Remarks to the Author:

I applaud the authors for their efforts in revising the paper. I particularly appreciate their effort to expand the analysis to other models, namely multiple variants of LLaMA2. I also appreciate their work on generating perturbations to the False Belief task, and the variants of the False Belief task.

I very much appreciate the points made by reviewer 2, and agree that one has to be careful before drawing very general conclusions, from the present analysis, about the true mental abilities of LLMs. But I think that the science of measurement of Theory of Mind itself has limitations that need to be considered when making an assessment. Indeed, I think the present paper can help motivate further work to produce more refined tasks for testing Theory of Mind. As such, the present paper brings a lot of value to the timely and important discussion of the LLM capabilities.

Reviewer #2:

Remarks to the Author:

I commend the authors for the significant work they've done from the previous manuscript to the current one. I recognize that it required a significant undertaking both for modeling and for collecting human data. I think this (together with the improved introduction and discussion throughout) have resulted in a stronger manuscript.

I'm sorry to say though that the changes have not changed my own view in terms of accepting the piece. If anything, since the original submission even more work has come out that rigorously 'stress tests' current LLMs for Theory-of-Mind (For example, "FANTOM: A Benchmark for Stress-testing Machine Theory of Mind in Interactions", I note that this work appeared before the re-submission) and arrives at rather negative conclusions regarding ToM in current LLMs. I know it's quite annoying that the LLM landscape is moving fast, and I recognize the authors are trying to do journal-work in a field where things keep getting published in conferences and arXiv, but that simply is the current landscape, and it seems equally problematic to publish a major journal piece about something that is a temporary snapshot which doesn't address the current concerns.

I won't belabor the authors by doing a line-by-line response to their line-by-line response, but I will note just a few things in passing (please understand that I don't mean this as "here are the major issues with the rebuttal and if you address these then everything is good to go, I mean this as 'there are many things I don't agree with and I am picking out a few as examples')

To comment R.2.2

"R2.2. The Theory of Mind tests presented in the battery, with the exception of the Irony comprehension test, are validated social psychological tests in humans. These have been shown to

have high internal validity and high reliability for measuring human social cognitive skills. These tests therefore provide a reliable benchmark for measuring LLMs against human performance, and using these validated tests is all the more important given recent claims that LLMs demonstrate a human-like capacity for Theory of Mind that rely on the use of single tests or even single items."

This comment seems to fundamentally misunderstand the problem with re-using existing tests from the literature when it comes to LLMs. Yes, things like the Sally-Ann task have high internal validity and high reliability, which make them a great test...for testing 4-year-old humans, who have not seen 10,000 of these examples and memorized them. If the fundamental issue at stake is whether LLMs are answering questions through memorizing statistics or through reasoning about mental states then it becomes very important not to re-use tests that have been mentioned in the scientific literature a billion times, or rather, it's ok to use them but it is the lowest bar to clear.

I'm sorry to repeat myself but since it apparently wasn't clear the first time maybe another example would help: It's perfectly valid to ask a child "what is $241 * 3728$ " as part of an on-the-spot pop-quiz, in order to see if they 'know how to multiply'. Even if we don't have access to their internal mental state, we are relatively secure in thinking they probably haven't seen that question before, and so if they arrive at the right answer, it is probably due to understanding some multiplication algorithm. We can further validate that intuition with various tests.

However, if my training set for a machine includes 10,000 copies of the text " $241 * 3728 = 898448$ ", and then I ask it " $241 * 3728 =$ " and it says "898448" then...sure, that's great. I would certainly be puzzled and worried and it if DIDN'T return 898448, but I am not yet convinced it used a multiplication. Then, when I try something not from the training like " $361 * 1261$ " and it gets it wrong, I am certainly justified in being more concerned.

=====

To the comment that people get some of the stress tests wrong too: This is interesting but isn't really dug into in the main text nor handled except to say that people get some of this wrong too. LLMs seem much worse than people at these, in a way that speaks against the central idea that they have ToM. To go back to the multiplication example, if your central argument is that your machine figured out how to multiply things, and then it utterly fails a bunch of novel multiplication questions that aren't in its training, it's a bad look to say "well...people get a bunch of multiplication questions wrong too". Sure, they do, but are we now talking about competence, or performance? How do they get it wrong? Is it in the same way as the machines? The whole thing is simply swept under the rug currently.

=====

to Comment R2.3:

"As discussed in Supplementary Section 4, it is worth noting that these control variants present diverse challenges for LLMs that go beyond tracking mental states, and involve understanding physical properties, relationships between objects, and spatial reasoning capabilities that have nothing to do with Theory of Mind."

This seems to miss the fact that the original ToM tasks also implicitly require an understanding of all those things. When Sally goes out of the room and Ann moves her box, we assume Sally can't see it because she's not in the room (understanding line of sight, physics, optics). When she comes back she doesn't immediately go for the new location, again this is based on understanding physical properties, spatial properties, and relationships between objects.

So, yes, those things are important, but they are present in the original tasks as well.

Author Rebuttal, first revision:

Reviewer #1

1. I applaud the authors for their efforts in revising the paper. I particularly appreciate their effort to expand the analysis to other models, namely multiple variants of LLaMA2. I also appreciate their work on generating perturbations to the False Belief task, and the variants of the False Belief task.

I very much appreciate the points made by reviewer 2, and agree that one has to be careful before drawing very general conclusions, from the present analysis, about the true mental abilities of LLMs. But I think that the science of measurement of Theory of Mind itself has limitations that need to be considered when making an assessment. Indeed, I think the present paper can help motivate further work to produce more refined tasks for testing Theory of Mind. As such, the present paper brings a lot of value to the timely and important discussion of the LLM capabilities.

***R1.1.** We thank the Reviewer for their time and effort in reviewing the manuscript and for their positive evaluation following constructive comments.*

Reviewer #2

1. I commend the authors for the significant work they've done from the previous manuscript to the current one. I recognize that it required a significant undertaking both for modeling and for collecting human data. I think this (together with the improved introduction and discussion throughout) have resulted in a stronger manuscript.

***R2.1.** We thank the Reviewer for their time and effort in reviewing the manuscript and appreciate their recognition of the enhancements made, leading to a more robust manuscript.*

2. I'm sorry to say though that the changes have not changed my own view in terms of accepting the piece. If anything, since the original submission even more work has come out that rigorously

'stress tests' current LLMs for Theory-of-Mind (For example, "FANTOM: A Benchmark for Stress-testing Machine Theory of Mind in Interactions", I note that this work appeared before the re-submission) and arrives at rather negative conclusions regarding ToM in current LLMs. I know it's quite annoying that the LLM landscape is moving fast, and I recognize the authors are trying to do journal-work in a field where things keep getting published in conferences and arXiv, but that simply is the current landscape, and it seems equally problematic to publish a major journal piece about something that is a temporary snapshot which doesn't address the current concerns.

R2.2. *We appreciate the Reviewer bringing this paper to our attention. It is indeed highly relevant and in line with our own intuitions that more naturalistic and interactive tools are required for evaluating the emergent capacities of LLMs. We have incorporated this citation into our revision. The Reviewer is correct that the LLM landscape is fast moving. From our perspective, this only reinforces the need of rigorous, systematic testing and proper validation in human samples as a fundamental foundation.*

3. I won't belabor the authors by doing a line-by-line response to their line-by-line response, but I will note just a few things in passing (please understand that I don't mean this as "here are the major issues with the rebuttal and if you address these then everything is good to go, I mean this as 'there are many things I don't agree with and I am picking out a few as examples')

To comment R.2.2

"R2.2. The Theory of Mind tests presented in the battery, with the exception of the Irony comprehension test, are validated social psychological tests in humans. These have been shown to have high internal validity and high reliability for measuring human social cognitive skills. These tests therefore provide a reliable benchmark for measuring LLMs against human performance, and using these validated tests is all the more important given recent claims that LLMs demonstrate a human-like capacity for Theory of Mind that rely on the use of single tests or even single items."

This comment seems to fundamentally misunderstand the problem with re-using existing tests from the literature when it comes to LLMs. Yes, thing like the Sally-Ann task has high internal validity and high reliability, which make them a great test...for testing 4-year-old humans, who have not seen 10,000 of these examples and memorized them. If the fundamental issue at stake is whether LLMs are answering questions through memorizing statistics or through reasoning about mental states then it becomes very important not to re-use tests that have been mentioned in the scientific literature a billion times, or rather, it's ok to use them but it is the lowest bar to clear.

I'm sorry to repeat myself but since it apparently wasn't clear the first time maybe another example would help: It's perfectly valid to ask a child "what is $241 * 3728$ " as part of an on-the-spot pop-quiz, in order to see if they 'know how to multiply'. Even if we don't have access to their internal mental state, we are relatively secure in thinking they probably haven't seen that question before, and so if they arrive at the right answer, it is probably due to understanding some multiplication algorithm. We can further validate that intuition with various tests.

However, if my training set for a machine includes 10,000 copies of the text " $241 * 3728 = 898448$ ", and then I ask it " $241 * 3728 =$ " and it says "898448" then...sure, that's great. I would certainly be puzzled and worried and it if DIDN'T return 898448, but I am not yet convinced it used a multiplication. Then, when I try something not from the training like " $361 * 1261$ " and it gets it wrong, I am certainly justified in being more concerned.

R2.3. We acknowledge the Reviewer's concerns and agree that stress testing is crucial to test the limits of LLMs' capacities. The Reviewer mentions the False Belief task as an example of a particular weak point, and we agree on the limitations of approaches using this one task in isolation for evaluating machine Theory of Mind. Our approach to move beyond these limitations was i) to test Theory of Mind on a battery of tasks, including one task (Irony) that was not publicly available for online scraping; ii) to test novel items; and iii) to test perturbations of the False Belief test and develop new variants of the Faux Pas that test for a Clever Hans-type effect.

4. To the comment that people get some of the stress tests wrong too: This is interesting but isn't really dug into in the main text nor handled except to say that people get some of this wrong too. LLMs seem much worse than people at these, in a way that speaks against the central idea that they have ToM. To go back to the multiplication example, if your central argument is that your machine figured out how to multiply things, and then it utterly fails a bunch of novel multiplication questions that aren't in its training, it's a bad look to say "well...people get a bunch of multiplication questions wrong too". Sure, they do, but are we now talking about competence, or performance? How do they get it wrong? Is it in the same way as the machines? The whole thing is simply swept under the rug currently.

R2.4. We agree with the Reviewer that this is a highly important question, and not one that is limited to incorrect answers: why do they get it wrong (or right)? Our motivation for developing the Faux Pas Likelihood Test and the Belief Likelihood Test was precisely to address this question. These tests were designed to reveal not just differences in performance - which was notably lower than humans for GPT-4 in the original Faux Pas Test and at ceiling for LLaMA2-70B - but also the underlying reasons for successes or failures. Through these modified

versions of the original Faux Pas test, we were able to show that: i) GPT responses integrate information to accurately interpret the speaker's mental state. Poor performance on the original Faux pas test was due to an excess of caution in answering the belief question rather than a failure of inference; ii) LLaMA2-70B's responses, on the other hand, fail to integrate information, raising the concern that LLaMA2-70B's perfect performance of on the original task may be an illusion of understanding.

5. to Comment R2.3:

"As discussed in Supplementary Section 4, it is worth noting that these control variants present diverse challenges for LLMs that go beyond tracking mental states, and involve understanding physical properties, relationships between objects, and spatial reasoning capabilities that have nothing to do with Theory of Mind."

This seems to miss the fact that the original ToM tasks also implicitly require an understanding of all those things. When Sally goes out of the room and Ann moves her box, we assume Sally can't see it because she's not in the room (understanding line of sight, physics, optics). When she comes back she doesn't immediately go for the new location, again this is based on understanding physical properties, spatial properties, and relationships between objects.

So, yes, those things are important, but they are present in the original tasks as well.

R2.5. *As the Reviewer has pointed out, ceiling performance on the classic Sally-Anne task raises the concern that LLMs are overtrained on this task. Hence, the importance of developing perturbations. Our results indicate that LLMs fail on perturbations of the Sally-Anne task adapted from Ullman (2023). Our point stands that this failure is difficult to interpret, firstly, because human participants also fail on half of these perturbations (see Supplementary Information, Section 4); secondly, because these perturbations also involve changes in the physical properties of the environment. LLMs might fail because they are sticking to the familiar script and are unable to automatically attribute an updated belief, or because they do not understand physical principles (e.g. transparency). We hope this clarifies our reasoning.*

Final Decision Letter:

Dear Dr Strachan,

We are pleased to inform you that your Article "Testing Theory of Mind in Large Language Models and Humans", has now been accepted for publication in Nature Human Behaviour.

Please note that *Nature Human Behaviour* is a Transformative Journal (TJ). Authors may publish their research with us through the traditional subscription access route or make their paper immediately open access through payment of an article-processing charge (APC). Authors will not be required to make a final decision about access to their article until it has been accepted. Find out more about Transformative Journals

We welcome the submission of potential cover material (including a short caption of around 40 words) related to your manuscript; suggestions should be sent to Nature Human Behaviour as electronic files (the image should be 300 dpi at 210 x 297 mm in either TIFF or JPEG format). Please note that such

pictures should be selected more for their aesthetic appeal than for their scientific content, and that colour images work better than black and white or grayscale images. Please do not try to design a cover with the Nature Human Behaviour logo etc., and please do not submit composites of images related to your work. I am sure you will understand that we cannot make any promise as to whether any of your suggestions might be selected for the cover of the journal.

With best regards,

[REDACTED]